# Molecular characterization of organic matter mobilized from Bangladeshi aquifer sediment: tracking carbon compositional change during microbial utilization

Lara E. Pracht[1], Malak M. Tfaily[2], Robert J. Ardissono[1], Rebecca B. Neumann[1]

[1]Department of Civil and Environmental Engineering, University of Washington, Seattle, 8195, USA
[2]Environmental Molecular Sciences Laboratory, Pacific Northwest National Laboratory, Richland, 99354, USA

*Correspondence to*: Rebecca B. Neumann (rbneum@uw.edu)

**Abstract.** Bioavailable organic carbon in aquifer-recharge waters and sediments can fuel microbial reactions with
implications for groundwater quality. A previous incubation experiment showed that sedimentary organic carbon (SOC) mobilized off sandy sediment collected from an arsenic-contaminated and methanogenic aquifer in Bangladesh was bioavailable; it was transformed into methane. We used high-resolution mass spectrometry to molecularly characterize this mobilized SOC, reference its composition against dissolved organic carbon (DOC) in aquifer recharge water, track compositional changes during incubation, and advance understanding of microbial processing of organic carbon in anaerobic
environments. Organic carbon mobilized off aquifer sediment (i.e., mobilized SOC) was more diverse, proportionately larger, more aromatic and more oxidized than DOC in surface recharge. It was predominately composed of terrestrially derived organic matter and had characteristics signifying it evaded microbial processing within the aquifer. Approximately 50% of identified compounds in mobilized SOC and in DOC from surface recharge water contained sulfur. During incubation, after mobilized SOC was converted into methane, new organosulfur compounds with high S-to-C ratios and high
nominal oxidation state of carbon (NOSC) were detected. We reason that these detected compounds formed abiotically following microbial reduction of sulfate to sulfide, which could have occurred during incubation but was not directly measured, or they were microbially synthesized. Most notably, microbes transformed all carbon types during incubation, including those currently considered thermodynamically unviable for microbes to degrade in anaerobic conditions (i.e., those with low NOSC). In anaerobic environments, energy yields from redox reactions are small and the amount of energy
required to remove electrons from highly reduced carbon substrates during oxidation decreases the thermodynamic favourability of degrading compounds with low NOSC. While all compound types were eventually degraded during incubation, NOSC and compound size controlled the rates of carbon transformation. Large more thermodynamically favourable compounds (e.g., aromatics with high NOSC) were targeted first while small less thermodynamically favourable compounds (e.g., alkanes and olefinics with low NOSC) were used last. These results indicate that in anaerobic conditions,
microbial communities are capable of degrading and mineralizing all forms of organic matter, converting larger energy-rich compounds into smaller energy-poor compounds. However, in an open system, where fresh carbon is continually supplied,

the slower degradation rate of reduced carbon compounds would enable this portion of the organic carbon pool to build up, explaining the apparent persistence of compounds with low NOSC in anaerobic environments.

# 1 Introduction

Organic carbon can impact groundwater quality both positively and negatively by fueling microbial reactions that remove and contribute dissolved contaminants to groundwater (e.g., natural attenuation of nitrate, mobilization of geogenic arsenic, methane production). The organic carbon fueling these microbial reactions can come from the near-surface environment, transported into the aquifer with recharging water (LeBlanc, 1984; Mailloux et al., 2013; Neumann et al., 2010), or can reside in the aquifer sediment, co-deposited when the aquifer formed (Korom, 1992; Parkin and Simpkins, 1995; Postma et al., 2007). Ultimately, the extent to which these sources of organic carbon can fuel subsurface reactions depends on their availability to the resident microbial community.

Historically, it was thought that molecular structure dictated carbon bioavailability (Lehmann and Kleber, 2015; Lutzow et al., 2006; Schmidt et al., 2011). Simple monomers were considered bioavailable, while more complex compounds, particularly those with aromatic rings, were considered molecularly recalcitrant. However, it is now understood that given the correct conditions, microbes can process all carbon types, continually transforming large, energy rich organic compounds into smaller, energy poor organic compounds (Lehmann and Kleber, 2015). In fact, compounds previously considered recalcitrant often have rapid turnover rates (Amelung et al., 2008).

The current view is that accessibility and microbial ecology control carbon bioavailability (Lehmann and Kleber, 2015; Schmidt et al., 2011). Physical protection mechanisms, such as organic carbon sorption to mineral surfaces and encapsulation into hydrophobic biopolymers or regions, can limit microbial access to carbon (Baldock and Skjemstad, 2000; Petridis et al., 2014), and energetic constraints, nutrient limitations, and metabolic capacities of microbial communities can affect the ability of microbes to process different chemical forms of organic carbon (Fontaine et al., 2007; LaRowe and Van Cappellen, 2011; Trulleyova and Rulik, 2004). Recent work has demonstrated that in anaerobic settings, the nominal oxidation state of carbon (NOSC) influences bioavailability. In a sulfate reducing aquifer, compounds with high NOSC were missing while those with low NOSC persisted (Boye et al., 2017). This outcome reflects the fact that organic carbon oxidation (i.e., removing electrons from carbon) requires energy (LaRowe and Van Cappellen, 2011), which must be offset by energy released from the reduction half-reaction. The amount of energy required to fully oxidize a carbon compound increases with the number of electrons. Thus, in anaerobic conditions, where the amount of energy released from reduction half-reactions is lower than that in oxic conditions, it is, paradoxically, more thermodynamically favorable to oxidize compounds with high NOSC than compounds with low NOSC (Keiluweit et al., 2016). It is these types of physical, energetic and nutritional constraints on carbon bioavailability that help explain the presence of old organic carbon (i.e., thousands of years old) within soil and sediments, and the contemporary processing of this organic carbon, particularly after perturbation events (Fontaine et al., 2007; Gurwick et al., 2008; Parkin and Simpkins, 1995).

Within aquifers, field studies have demonstrated that the reactivity of sedimentary organic carbon (SOC) decreases with increasing age (Chapelle et al., 2009; Jakobsen and Postma, 1994; Postma et al., 2012). With contemporary understanding of the factors that control carbon bioavailablity, this diminishment in reactivity can be attributed to the development of constraints limiting microbial processing of sedimentary organic carbon rather than to a complete absence of SOC. Results from an incubation experiment conducted with sediments collected from a methanogenic, alluvial aquifer in Munshiganj, Bangladesh align with such an interpretation (Neumann et al., 2014). Sampling and homogenization of the aquifer sediments mobilized $0.33 \pm 0.06$ mg-C of organic carbon per gram of sediment or $8.8 \pm 0.7\%$ of total SOC (SI section 1). During incubation, the mobilized SOC was rapidly converted into methane by the native microbial community (Neumann et al., 2014). The experiment demonstrated that the ~5000 year-old, sandy aquifer sediments contain a notable amount of organic carbon that, if mobilized within the aquifer, could fuel microbial reactions.

In the Neumann et al. (2014) incubation experiment, SOC was mobilized during sampling and/or homogenization of the aquifer sediment. While this type of physical disturbance to the aquifer matrix could not occur *in situ* (i.e., within the aquifer), it is plausible that more realistic aquifer perturbations, such as geochemical changes brought about by large-scale groundwater pumping, could mobilize SOC *in situ*. For example, mineral-associated organic carbon can get mobilized into groundwater if solution pH increases (Gu et al., 1994a; Jardine et al., 1989; Kaiser and Zech, 1999), if concentrations of ions that compete with organic carbon for sorption sites increase (Jardine et al., 1989; Kaiser and Zech, 1999), or if an influx of dissolved organic carbon fuels microbial reactions that target the mineral phase (e.g., reductive dissolution of iron oxide minerals) (Eusterhues et al., 2003; Fontaine et al., 2007; Mikutta et al., 2006).

At two different sites in Asia, there is published evidence that large-scale groundwater pumping has facilitated mobilization of sedimentary organic carbon. At the site in Bangladesh where materials for the Neumann et al. (2014) incubation were collected, dry-season groundwater irrigation has pulled surface recharge water down to a 30-m depth where irrigation wells are screened (Harvey et al., 2006). Field measurements showed that dissolved organic carbon (DOC) concentrations in groundwater remained constant down to this depth, but that DOC age increased (Harvey et al., 2002). This situation is only possible if SOC was released along the groundwater flow path as younger DOC was mineralized or sequestered into sediment (Neumann et al., 2010). In Vietnam, urban groundwater pumping for the city of Hanoi has pulled organic-rich river water into a Pleistocene-aged aquifer (van Geen et al., 2013). Along the recharge pathway, groundwater DOC concentrations peaked where microbial reduction reactions were occurring (i.e., iron reduction and arsenic mobilization), and mass balance calculations demonstrated that the river could not have supplied enough carbon to fuel these reactions. The data indicated that DOC in river water fuelled microbial reactions that mobilized SOC and that the mobilized SOC then fueled further reduction reactions.

Given the potential importance of SOC for fueling subsurface reactions, we sought to chemically characterize the carbon compounds mobilized off the aquifer sediment in the Neumann et al. (2014) incubation. While mobilization was an artifact of the experiment, it allowed us to study a pool of sedimentary organic carbon that was highly bioavailable after mobilization and would have otherwise been inaccessible. For characterization, we used Fourier transform ion cyclotron resonance mass spectrometry (FT-ICR-MS) coupled with electrospray ionization (ESI) in negative ion-mode. FT-ICR-MS is

capable of characterizing the thousands of different compounds contained in natural organic carbon; though the technique is not quantitative because the ionization efficiency of compounds vary, making some compounds more difficult or impossible to identify (Sleighter and Hatcher, 2007). The technique has been applied to natural organic matter in a variety of environments (Chipman et al., 2010; Evert, 2015; Kim et al., 2006; Longnecker and Kujawinski, 2011; Spencer et al., 2014; Tfaily et al., 2013).

Through FT-ICR-MS analysis of the Neumann et al. (2014) experimental samples, we were not only able to chemically characterize SOC mobilized from the Bangladeshi aquifer sediments, but were also able to compare the chemical composition of mobilized SOC to that of DOC in surface-water sources that recharge the aquifer and to track compositional changes in mobilized SOC as it was converted into methane by the native microbial community during incubation. These efforts generate knowledge about the chemistry of organic carbon within surface recharge waters and sediments of the Bangladeshi alluvial aquifer — the two possible sources of organic carbon for fueling subsurface microbial reactions — and advance understanding of microbial processing of organic carbon within anaerobic environments.

## 2 Material and Methods

### 2.1 Incubation

FT-ICR-MS samples were collected from the Neumann et al. (2014) incubation experiment, which used aquifer sediment and aquifer recharge waters collected from a field site in Munshiganj, Bangladesh. The field site has been well characterized hydrologically and geochemically (Harvey et al., 2002, 2006; Swartz et al., 2004). The site is overlain by 3.5 m thick clay layer that, at deeper depths, remains saturated year round. This clay layer hosts rice fields and man-made ponds that serve as the two primary recharge sources for the upper aquifer (Harvey et al., 2006). The upper aquifer, in turn, supplies irrigation water for dry-season rice production (Figure 1). This aquifer is ~5,000 years old and contains Holocene deposits composed of gray colored fine and silty sands between 3.5 m and 12 m depths, and grayish-green colored fine to medium sands between 12 m and 110 m depths (Swartz et al., 2004). The entire aquifer is anaerobic. Groundwater at the site is contaminated with arsenic and supersaturated with methane (Harvey et al., 2002).

Neumann et al. (2014) collected sediment for the incubation experiment from an aquifer depth of 9.1 m (Figure 1), where groundwater arsenic and methane concentrations were relatively low (~6 $\mu$g-As L$^{-1}$ and ~5 mg-CH$_4$ L$^{-1}$, respectively). They homogenized the sediment and collected sediment porewater (i.e., water surrounding the aquifer sediment) by vacuum filtering (0.2 $\mu$m) multiple sediment aliquots. Dissolved organic carbon (DOC) concentrations in sediment porewater ranged between 856 and 1219 mg-C L$^{-1}$ (Neumann et al., 2014) (SI Table S1), which was much higher than ~4 mg-C L$^{-1}$ DOC concentration measured in groundwater at the field site (Swartz et al., 2004). Neumann et al. (2014) concluded that organic carbon was mobilized off the sediment into porewater during sampling and/or homogenization of the sediment. In this manuscript, we refer to organic carbon dissolved in the Neumann et al. (2014) sediment porewater as 'mobilized SOC.'

After homogenization, Neumann et al. (2014) dried and sterilized by gamma irradiation half of the sediment. They independently incubated both native and sterilized sediment for 200+ days with filtered (0.45 μm) aquifer recharge water that they had collected from a rice field and pond at the Munshiganj field site (8.2 g dry sediment to 10 mL of water). They collected the rice field recharge water from a shallow well installed within a rice field bund (i.e., the raised boundary surrounding a rice field), screened just below the surficial clay layer hosting the rice field (Figure 1); rice field recharge is focused through bunds. They collected pond recharge water by driving a Push Point Harpoon (M.H.E. products) 44 cm into the bottom sediments of a pond (Figure 1). In this manuscript, we refer to DOC in the rice field and pond recharge waters as 'recharge DOC.'

Neumann et al. (2014) destructively sampled incubation bottles 1, 17–20, 80–81, 91, 184 and 273 days after initiation of the experiment (Neumann et al., 2014). They vacuum filtered (0.2 μm) incubation water and analyzed it for dissolved organic carbon. They also analyzed headspaces from a few incubation bottles for methane. Previously published (Neumann et al., 2014) DOC and methane concentrations for all incubation replicates and time points are presented in SI Tables S2–S3. During processing of the incubation bottles, aliquots (100–200 μL) of filtered water were saved and immediately frozen at –18 °C. These frozen subsamples were later analyzed by FT-ICR-MS (see next section). Results from the FT-ICR-MS analysis are the focus of this manuscript.

Neumann et al. (2014) kept all water and sediment samples anoxic during collection, handling and processing steps through the use of an anerobic chamber, gas-impermeable bags, oxygen scavenging packets with oxygen indictor tabs, glass BOD bottles, and glass serum vials sealed with butyl rubber stoppers. Further sampling details are available in the Neumann et al. (2014) manuscript.

## 2.2 FT-ICR-MS analysis

Two to three months after collection, frozen aliquots of water from the Neumann et al. (2014) incubation experiment were packed in dry ice for shipment to the Environmental Molecular Sciences Laboratory (EMSL) at Pacific Northwest National Laboratory in Richland, WA. At EMSL samples were diluted 1:2 (v/v) with LC-MS grade methanol (MeOH) less than 30 minutes before analysis to minimize esterification (McIntyre and McRae, 2005), and were injected directly into the instrument. Solid phase extraction was not performed due to sample volume constraints. No salts were observed in the spectra. To avoid biases in ionization efficiency due to differences in DOC concentrations, the ion accumulation time (IAT) was optimized for each sample individually based on previously measured DOC concentrations (Tables S1 and S2). The IAT ranged between 1-1.5s. All samples were run with instrument settings optimized by tuning on the Suwannee river fulvic acid (SRFA) standard.

A standard Bruker ESI source was used to generate negatively charged molecular ions. Negative ion mode was chosen due to its extensive use in characterizing a broad range of environmental DOC samples. Additionally, previous work has shown that organic matter is rich in carboxylic groups, as these groups ionize best in negative mode (Stenson et al.,

2003). Recently, Ohno et al. (2016) have shown that for samples rich in condensed aromatic and aromatic molecules, the use of (+)ESI improves ionization of aliphatic and carbohydrate like SOM components. Because we did not complement (-)ESI with (+)ESI, our study may underestimate the presence of aliphatic and carbohydrate molecules.

Samples were introduced to the ESI source equipped with a fused silica tube (200 μm i.d) through a syringe pump at a flow rate of 3.0 μL/min. Experimental conditions were as follows: needle volt-age, +4.4 kV; Q1 set to 150 m/z; and the heated resistively coated glass capillary operated at 180 °C. Blanks (HPLC grade MeOH) were run at the beginning and end of the analysis to monitor potential carry over from one sample to another. A series of three free fatty acids with masses around 255, 283 and 325 Da were observed in the solvent blanks. These peaks did not show up when running the samples; analytes in the samples picked up the charges and had higher intensity than the three fatty acids present in the MeOH. The instrument was flushed between samples with a mixture of water and methanol. The instrument was externally calibrated weekly with a tuning solution from Agilent, which calibrates to a mass accuracy of <0.1. ppm and contains the following compounds: $C_2F_3O_2$, $C_6HF_9N_3O$, $C_{12}HF_{21}N_3O$, $C_{20}H_{18}F_{27}N_3O_8P_3$, and $C_{26}H_{18}F_{39}N_3O_8P_3$ with an m/z ranging between 112 to 1333 Da.

One hundred forty four individual scans were averaged for each sample and internally calibrated using OM homologous series separated by 14 Da (–CH2 groups). The mass measurement accuracy was 0.4 ppm on average and no larger than 1 ppm for singly charged ions across a broad m/z range (i.e. 200 <m/z <1200). Data analysis software (BrukerDaltonik version 4.2) was used to convert raw spectra to a list of m/z values applying FTMS peak picker with S/N threshold set to 7 and absolute intensity threshold set to the default value of 100. Elemental formulas were assigned using in-house built software following the Compound Identification Algorithm (CIA), described by Kujawinski and Behn (2006) and modified by Minor et al. (2012). Elemental formulas were assigned based on the following criteria: S/N >7, and mass measurement error <1 ppm, taking into consideration the presence of C, H, O, N, S and P and excluding other elements. Formulas were allowed to have the following number of elements: C (1–100 atoms), H (1–200 atoms), O (1-30 atoms), N (0–20 atoms), S (0–10 atoms), and P (0–6 atoms). Additionally, each phosphorus atom required at least four oxygen atoms. CH2, H2, NH and O were used for propagation. Since molecules containing both phosphorus and sulfur are rare, the elemental formula with the lowest error and the lowest number of heteroatoms was consistently chosen. All elemental formulas with errors above 1ppm were rejected.

To further reduce cumulative errors, all sample peak lists within the entire dataset were aligned to each other prior to elemental formula assignment to eliminate possible mass shifts that would impact assignment. Peaks with large mass ratios (m/z values >500 Da) often have multiple possible elemental formulas. These peaks were assigned formulas through the detection of homologous series (CH2, O, H2, NH). Specifically, when the m/z of a homologous series group and the m/z of an already confidently assigned compound were summed to an m/z that was observed >500 Da, the elemental formula assigned to the smaller compound was appended by the atoms of the homologous series group and the new elemental formula was assigned to the larger compound. If no elemental formula matched an m/z value within the allowed error, the peak was not included in the list of elemental formulas. 43 to 47% of mass spectra data were successfully assigned. Many

peaks were left unassigned because they appeared to be organo-metal complexes (potentially arsenic) and metals were not included in the formula assignment (inclusion of metal is not straightforward and requires further validation). Overall, the percentage assignment was consistent for all of the samples and was considered at a good level given the samples were not concentrated or otherwise treated prior to analysis.

## 2.3 Data processing

Due to variation in ionization efficiency of different compounds, peak presence was used for analysis of FT-ICR-MS data rather than peak intensity. Since low DOC concentrations can reduce ionization effectiveness, experimental replicates were combined during data processing, as were incubation samples with similar DOC concentrations collected in close temporal proximity to each other. Tables S1 and S2 indicate the number and DOC concentrations of samples combined together. Peaks present in any individual sample were considered present for the combined sample. Analysis focused only on combined samples, which included the rice field and pond recharge water, sediment porewater, and incubation water from day 1, days 17–20 (called day 18 throughout the manuscript), and days 80–91 (called day 85 throughout the manuscript). Combining replicates and samples with similar DOC concentrations improved robustness of sample characterization and minimized artifactual variability when looking at compositional changes across incubation treatments and across incubation time.

Elemental formulas in combined samples were used to divide organic carbon into 4 distinct heteroatom groups: CHO, CHON, CHO plus P (any compound with P and without S), and CHO plus S (any compound with S). Elemental formulas were also used to calculate double bond equivalent (DBE), nominal oxidation state of carbon (NOSC), aromaticity index (AI), and compound classification. For an organic compound $C_cH_hN_nO_oP_pS_s$, DBE was calculated as (Koch and Dittmar, 2006, 2016):

$$DBE = 1 + \frac{1}{2}(2c - h + n + p) . \tag{1}$$

NOSC was calculated as (LaRowe and Van Cappellen, 2011):

$$NOSC = 4 - (4c + h - 3n - 2o + 5p - 2s)/c . \tag{2}$$

AI was calculated as (Koch and Dittmar, 2006, 2016):

$$AI = \left(1 + c - 0.5 * o - s - 0.5 * (n + p + h)\right)/(c - 0.5 * o - s - n - p), \tag{3}$$

which identifies samples as alkanes (AI=0), olefinics (0<AI≤0.5), aromatics (AI>0.5), or condensed aromatics(AI≥0.67) (Koch and Dittmar, 2006, 2016; Willoughby et al., 2014). Compound class was assigned based on oxygen-to-carbon and hydrogen-to-carbon ratios (i.e., location within van Krevelen plots) (D'Andrilli et al., 2015; Kim et al., 2003). Table S4 includes the ratio ranges used to identify compounds that were lipid-, protein-, carbohydrate-, amino sugar-, lignin-, tannin-, and condensed hydrocarbon-like. Compounds falling outside of these ranges were assigned as 'other.'

Compound changes were tracked between day 1 and day 18 and between day 18 and day 85 of the incubation. Compounds identified at both time points were classified as 'common,' those identified at the first time point but not at the

second were classified as 'lost,' and those identified at the second time point but not at the first were classified as 'newly detected.'

## 3 Results and Discussion

### 3.1 Composition of DOC in aquifer recharge waters versus mobilized SOC

Microbial reactions within aquifers are fueled either by sedimentary organic carbon or by dissolved organic carbon transported from the surface into the aquifer with recharging water. Here we compare the chemical composition of these two sources of organic carbon for the Bangladeshi aquifer. As previously mentioned, ESI FT-ICR-MS is not quantitative because the ionization efficiency of the different organic compounds vary widely during ESI. Thus, comparisons involve only those compounds detected by the instrument, and not the full composition of each sample. However, all samples were treated the

same and each sample was normalized by the total number of detected peaks, making such comparisons more robust.

The chemical composition of DOC in the pond and rice field recharge waters collected for the Neumann et al. (2014) incubation experiment was highly similar (Figure 2, Table 1). Notable differences between DOC in the two recharge waters only existed in the H-to-C ratio distribution. Pond recharge contained proportionately fewer compounds with H-to-C ratios <0.75 and proportionately more compounds with H-to-C ratios >1.25 compared to rice field recharge (Figure 2b).

However, these ratio differences did not result in markedly different compound classifications (Figure 2h; Table 1).

In contrast, the character of organic carbon mobilized off aquifer sediment into sediment porewater during sampling and homogenization of the sediment was distinctly different than DOC in the two recharge waters. Sediment porewater, which was shown by Neumann et al. (2014) to have a higher dissolved organic carbon concentration than recharge waters ($1059 \pm 186$ mg-C $L^{-1}$ vs. $17 \pm 7$ mg-C $L^{-1}$ in rice field recharge water and $30 \pm 3$ mg-C $L^{-1}$ in pond recharge water), also had

a higher total number of assigned formulas (5263 vs. 627 in rice field recharge water and 835 in pond recharge water). The number of detected compounds provides insight into the complexity of the samples (Sleighter and Hatcher, 2008). These numbers indicate that mobilized SOC was more complex than DOC in surface recharge water. A recent study using FT-ICR-MS to probe the character of organic carbon in intact soil cores found that soluble carbon in the smaller, less physically accessible soil pores was more complex than soluble carbon in the larger, more physically accessible soil pores (Bailey et al.,

2017). In this study and in our study, the more complex carbon pool was the one that, due to *in situ* constraints (physical, nutritional and/or energetic), evaded microbial decomposition.

The chemical character of the mobilized SOC and its compositional difference from DOC in recharge water also signify that it avoided microbial processing within the aquifer. The mass distribution for mobilized SOC was wider and skewed toward larger masses relative to DOC in recharge (Figure 2d). The wider distribution aligns with the larger number,

and thus, greater diversity of compounds detected in mobilized SOC relative to DOC in recharge. The skew toward larger masses supports the idea that mobilized SOC was protected from *in situ* decomposition. Microbial communities continually transform large compounds into smaller compounds, and the breakdown of large macromolecules is the first step. During the

Neumann et al. (2014) incubation, large compounds were lost first while small compounds persisted (see section *3.3* and *3.4*). Additionally, the dimensions of bacterial porin structures can lead to the exclusion of large organic solutes (e.g. >600Da), requiring extracellular enzymes (exoenzymes) to breakdown these larger molecules before they can be assimilated by microbial cells (Benz and Bauer, 1988; Nikaido and Vaara, 1985; Weiss et al., 1991). Relative to DOC in recharge water, mobilized SOC contained proportionately more compounds that were larger than 600 Da (Table 1), implying that these compounds had not yet been attacked by exoenzymes.

Compositionally, mobilized SOC was more aromatic than recharge DOC. Similar to the mass distribution, the DBE distribution for mobilized SOC was wider and skewed toward higher numbers relative to DOC in recharge (Figure 2e), signifying SOC was composed of a wider diversity of compounds, including those with more double bonds and rings. Based on aromaticity index (AI), aromatic compounds accounted for the larger DBE. Mobilized SOC had proportionately fewer olefinic compounds (i.e., those with double bonds) but more aromatic compounds than DOC in recharge water (Figure 2g; Table 1). According to compound classifications, these aromatic compounds were tannin- and condensed hydrocarbon-like (Figure 2h; Table 1). Mobilized SOC also had a smaller proportion of compounds classified as protein- and lipid-like (Figure 2h; Table 1). These differences in compound classification imply that relative to DOC in recharge, mobilized SOC was composed of proportionately more terrestrially derived organic matter (i.e., cellulose, lignin and tannin-like compounds; ~29% versus ~22% in recharge), more highly condensed organic matter (i.e., coal-, soot-, charcoal- and black-carbon-like compounds; ~38% versus ~25% in recharge), and notably less microbially derived organic matter (i.e., lipid-, protein-, amino-sugar-like compounds; ~13% versus ~30% in recharge). The remaining proportion of compounds did not overlap with a known class.

The detected composition of the mobilized SOC signifies it was predominately composed of terrestrial organic matter that had not yet been processed by microbes, and that to some extent had experienced combustion or photochemical reactions (i.e., the highly condensed fraction) (Chen et al., 2014). Similarly, Bailey et al. (2017) found that soluble carbon in smaller, less accessible soil pores had proportionately more lignin-, tannin- and condensed hydrocarbon-like compounds and proportionately fewer lipid-like compounds than soluble carbon in larger, more accessible soil pores. Traditionally, lignins, tannins and condensed hydrocarbons were considered molecularly recalcitrant (Lehmann and Kleber, 2015; Lutzow et al., 2006; Schmidt et al., 2011), and thus their presence was historically interpreted as resulting from negative enrichment (i.e., microbes preferentially decomposing other compounds). However, in both the Bailey et al. (2017) study and in the Neumann et al. (2014) incubation (see section *3.4*), these terrestrially derived and highly condensed organic compounds were readily degraded. Thus, the presence of these compounds indicates instead that they were protected from *in situ* microbial attack, an interpretation that aligns with contemporary understanding that physical and ecological factors control carbon bioavailability rather than molecular structure.

Of all the compound classes presented in Figure 2h, tannin- and condensed hydrocarbon-like compounds have the highest nominal carbon oxidation state (NOSC) while lipid- and protein-like compounds have the lowest NOSC (Boye et al., 2017; Keiluweit et al., 2016). The relative enrichment of tannin- and condensed hydrocarbon-like compounds and depletion

of lipid- and protein-like compounds in mobilized SOC (Figure 2h) explains the positive skew of the NOSC distribution for mobilized SOC relative to recharge DOC (Figure 2f). The median NOSC value for mobilized SOC was +0.10 while the median value for DOC in pond and rice field recharge was –0.56 and –0.32, respectively (Table 1). Not surprisingly, Bailey et al. (2017) similarly found that soluble carbon from the smaller, less accessible soil pores had a higher NOSC than soluble carbon from the larger, more accessible soil pores, though the difference was not statistically significant. In the anaerobic conditions of the Bangladeshi aquifer, where highly energetic electron acceptors like oxygen and nitrate are absent (Swartz et al., 2004), it is more thermodynamically favorable to oxidize compounds with high NOSC (Keiluweit et al., 2016; LaRowe and Van Cappellen, 2011). Thus, the presence of these more oxidized, thermodynamically favorable carbon compounds within mobilized SOC further supports the idea that this carbon pool evaded microbial processing within the aquifer.

Notably, in mobilized SOC and in recharge DOC, roughly half of all identified molecules had sulfur heteroatoms (47% in pond recharge, 59% in rice field recharge, and 51% in SOC) (Figure 2a; Table 1). Organosulfur compounds form in organic, sulfate-reducing environments where reduced sulfide species can react with various types of organic matter (Heitmann and Blodau, 2006; Perlinger et al., 2002; Schouten et al., 1993). As such, organosulfur compounds have been detected in various anaerobic environments (Brown, 1986; Urban et al., 1999). In an aquifer system in Rifle, CO, sulfur heteroatom abundance increased as DOC was sampled from more chemically reduced zones (Evert, 2015). Sediment and water samples obtained for the Neumann et al. (2014) incubation study similarly came from chemically reduced environments. Groundwater at the depth from which the sediment was collected contained ~7 mg-S $L^{-1}$ as sulfate and 64 mg-S $L^{-1}$ as sulfide (Swartz et al., 2004). Sulfurization of organic matter is an abiotic process that occurs early in sediment diagenesis at ambient temperatures and pressures (Kohnen et al., 1991; Schouten et al., 1993). However, biologically-mediated reactions can also involve organosulfur compounds (Brosnan and Brosnan, 2006; Madigan et al., 2003; Thauer, 1998). In the later incubation period, new organosulfur compounds were detected in biotic incubations but not in abiotic incubations, indicating that microbes either generated sulfide (via dissimilatory sulfate reduction) that abiotically reacted with DOM during this time period or they directly synthesized organosulfur compounds (see section *3.4*).

**3.3 Carbon transformations between day 1 and day 18 of incubation**

At the onset of the incubation experiment, 1 day after recharge waters were mixed with aquifer sediment, DOC concentration and composition were dominated by sediment porewater. Average DOC concentrations were 140-300 mg-C $L^{-1}$ (Figures 3a-d; Table S2), which Neumann et al. (2014) showed were explained by dilution of mobilized SOC in sediment porewater (1059 ± 186 mg-C $L^{-1}$) with recharge water (17 ± 7 mg-C $L^{-1}$ and 30 ± 3 mg-C $L^{-1}$, in rice field and pond recharge, respectively). Characterization by FT-ICR-MS showed that DOC in these initial incubation samples was more similar to mobilized SOC in sediment porewater than to DOC in aquifer recharge water (Figure S1). Both mass balance and chemical characterization indicate that the primary source of soluble organic carbon in the incubation experiment was that mobilized from the aquifer sediment.

Between day 1 and day 18, average DOC concentrations increased by 13±5 to 28±16 mg-C L$^{-1}$ in incubations conducted with pond recharge (Figures 2a-b; Table S2) and average concentrations did not noticeably change in incubations conducted with rice field recharge, after accounting for variability between replicates (Figures 2c–d; Table S2). During this time period, the number of compounds identified by FT-ICR-MS increased in all treatments. In the two pond incubations ~4000 new compounds were detected (Figures 2e–f), and in the two rice field incubations ~2000 new compounds were detected (Figures 2g-h). Across all four treatments, the compositional character of newly detected compounds on incubation day 18 matched that of DOC in initial incubation water (i.e., from day 1), which, in turn, was similar to SOC mobilized into sediment porewater (SI Figure S2). This correspondence can be explained by abiotic desorption of SOC from aquifer sediment into incubation water between day 1 and day 18 of the incubation, with newly detected compounds representing those desorbed from aquifer sediment. In support of abiotic desorption, during this time period there was a proportionately greater detection of large compounds (Figure 4, SI Figure S3, SI Figure S4), which can be explained by slower diffusive partitioning rates for larger compounds. If this interpretation is correct, then increases in DOC concentrations between day 1 and day 18 of the incubation can be attributed to desorption of SOC.

Fewer compounds were lost between day 1 and day 18 than were newly detected, and the type of recharge water used in incubation had no impact on the number of lost compounds (Figure 2e–h). Instead, use of sterilized versus native sediment controlled compound loss. Treatments with native sediment lost ~1500 compounds (Figures 2f,h) while treatments with sterilized sediment lost ~800 compounds (Figures 2e,g). Greater loss of compounds in incubation sets conducted with native sediment indicates that microbial transformation of organic carbon occurred during this early incubation period. Figures 4 and S3 show that compounds in the CHO heteroatom group with high O-to-C ratios and high NOSC were lost in proportionately greater numbers in biotic than in abiotic incubations. These compounds had lower H-to-C ratios, were predominately smaller than 600 Da, and were identified as aromatic (Figures 4 and S3).

The preferential loss of aromatic compounds observed in the biotic anaerobic incubation (Figures 4 and S3) contrasts with that typically observed in aerobic incubations. In aerobic conditions, microbes do not initially target aromatic compounds. While aromatic compounds eventually get oxidized, in the initial phases of aerobic degradation, they are actually selectively preserved (Lutzow et al., 2006; Schmidt et al., 2011). In both aerobic and anaerobic conditions, the rate of carbon oxidation depends both on the thermodynamic driving force driving the reaction (i.e., the net difference between energy gained from the reaction and energy needed to synthesize ATP) and on the ability of microbial communities to acquire and process reactants (Jin and Bethke, 2003). When oxygen is the electron acceptor, the thermodynamic driving force of carbon oxidation does not depend on NOSC (Keiluweit et al., 2016). Thus, no carbon form is more or less energetically favorable to oxidize. Instead, non-thermodynamic factors, such as enzyme kinetics, influence the rate of microbial utilization. In contrast, with anaerobic electron acceptors, the thermodynamic driving force rapidly drops as NOSC decreases (Keiluweit et al., 2016). Thus, in anaerobic environments, compounds with high NOSC represent the most energetically favorable compounds to oxidize. These energy dynamics explain why aromatic compounds with high NOSC and high O-to-C ratios were initially targeted in the anaerobic incubations. Additionally, the <600Da size means these

compounds had a high likelihood of being directly assimilated by microbes (Benz and Bauer, 1988; Nikaido and Vaara, 1985; Weiss et al., 1991).

**3.4 Carbon transformations between day 18 and day 85 of incubation**

Between day 18 and 85, DOC was lost from incubations conducted with native sediment (Figure 3b,d), but not from those conducted with sterilized sediment (Figure 3a,c). Carbon mass balance calculations conducted by Neumann et al. (2014) indicated that DOC lost in the biotic incubations was transformed into methane. Accordingly, the number of identified compounds by FT-ICR-MS decreased in both of the biotic incubations (Figure 3f,h) but did not change in the abiotic incubations (Figure 3e,g). However, there was not a direct relationship between DOC concentration and number of compounds; both biotic incubations lost similar amounts of DOC by day 85 (Figure 3b,d; Table S2), but the decrease in the number of compounds was larger in incubations conducted with pond recharge than in those conducted with rice field recharge (Figure 3f,h).

All compound types were microbially transformed, as evidenced by proportionally greater compound changes in biotic than in abiotic incubations between day 18 and 85 of the incubation, regardless of chemical index (Figures 5 and S5). Compounds traditionally considered molecularly recalcitrant (e.g., aromatics and condensed aromatics) were almost fully transformed in the biotic incubations, reinforcing current understanding that microbial communities can decompose a wide range of carbon types (Lehmann and Kleber, 2015; Schmidt et al., 2011). Additionally, compounds with low NOSC were lost from the biotic incubations (Figure 5 and S5). The loss of compounds with low NOSC in an anaerobic incubation conflicts with the recently presented idea that microbial degradation of these compound types is thermodynamically unviable in anaerobic environments (Boye et al., 2017; Keiluweit et al., 2016). The argument assumes that the thermodynamic driving force (i.e., net difference between energy gained from the reaction and energy needed to synthesize ATP) for oxidation of highly reduced carbon substrates approaches zero as less energetically rich, anaerobic electron acceptors get used in the redox reaction (Boye et al., 2017; Keiluweit et al., 2016). Boye et al. (2017) predicted that, in sulfidic conditions, compounds with NOSC below –0.3 are energetically unviable for respiring microbes. However, in the highly reducing environment of the incubation, where methanogenesis occurred, compounds with NOSC values as low as –2 were lost between incubation day 18 and day 85 (Figure 5, S5). This result indicates that microbial processing of these highly reduced compounds was not thermodynamically inhibited in the incubation despite the low energy yield associated with methanogenesis.

While all compound types were microbially transformed (i.e., were lost in greater proportion in the biotic than abiotic incubations), some compound types within the biotic incubations persisted between day 18 to day 85 (i.e., were common to both time points). Compound persistence was proportionately greater in the rice field incubation than in the pond incubation (Figures 5, S5 and S6). Our interpretation is that persistent compounds were not necessarily biologically inaccessible, but rather they were more slowly processed or were regenerated during degradation of larger compounds. One set of persistent compounds were those in the CHO heteroatom group with high H-to-C ratios, low NOSC, and small size (Figures 5, S5, S6). They were indexed as alkanes and olefinics. The persistence of CHO compounds with low NOSC

directly contrasts with the proportionately greater loss of CHO compounds with high NOSC in the early incubation phase (Figures 4 and S3). While NOSC did not control thermodynamic viability in the incubation, it did affect relative rates of compound transformation, with higher NOSC facilitating faster transformation, but only for CHO compounds. In other heteroatom groups, NOSC was not related to persistence (Figures 5 and S5). Instead, compound size predicted persistence, with the smallest compounds persisting in all groups (Figures 5, S5 and S6). While historically it was thought that larger, humified compounds resisted degradation, persistence of smaller compounds supports emerging understanding that organic carbon in the environment is continuously processed from larger to smaller molecular size (Lehmann and Kleber, 2015).

In contrast to persistence of smaller compounds, large compounds were lost and newly detected in both biotic and abiotic treatments in proportionately higher numbers than smaller compounds between day 18 and 85 (Figures 5, S5 and S6). These compound changes could reflect continued abiotic partitioning of large molecules between sediment and water, or, if the abiotic incubations did not stay completely sterile, it could reflect microbial processing of large compounds. However, if microbial processing occurred in the incubation conducted with sterilized sediment, it was slower than in biotic incubations and it did not alter DOC concentrations (Figure 3a,c).

A notable difference between incubations conducted with native and sterilized sediment was detection of new compounds in the CHO+S heteroatom group between day 18 and day 85 in native-sediment incubations (Figures 5, S5). The new CHO+S compounds had low H-to-C ratios, high S-to-C ratios, and high NOSC. They had a range of O-to-C ratios and spanned all aromaticity indices. Other studies have directly connected presence of organosulfur species with microbial activity (D'Andrilli et al., 2013; Gonsior et al., 2011). We reason that these detected compounds either formed due to microbial sulfate reduction and sulfide generation during this period, which was not directly measured, followed by abiotic sulfurization of organic matter, or they were microbally synthesized. Both pathways are plausible. Sulfurization of organic matter is well documented (Brown, 1986; Heitmann and Blodau, 2006; Kohnen et al., 1991; Urban et al., 1999), and microbes can synthesize a wide array of organic sulfur compounds (Brosnan and Brosnan, 2006; Madigan et al., 2003; Thauer, 1998).

## 4 Conclusions

### 4.1 Sedimentary organic carbon

Collectively, FT-ICR-MS characterization of SOC mobilized off the aquifer sediment in the Neumann et al. (2014) incubation indicates it was a highly diverse and energetically favorable pool of organic carbon, predominately composed of terrestrial- and fire-derived organic matter not yet processed by microbes (Figure 2). Similarities in the compositional character between this pool of SOC and soluble carbon from smaller, less physically accessible pores in an intact soil core study (Bailey et al., 2017) imply that when organic carbon evades microbial attack, it remains highly diverse and composed of more oxidized, aromatic compounds (e.g., tannins and condensed hydrocarbons) relative to carbon that is readily accessible to microbial processing (e.g, DOC in recharge water or soluble carbon in larger soil pores).

Roughly half of the identified compounds in mobilized SOC (and in recharge DOC) contained sulfur (Figure 2). During incubation, after mobilized SOC was transformed into methane, new sulfur-containing compounds were detected in biotic incubations that had high S-to-C ratios and high NOSC values (Figures 5 and S5). These compounds were not detected in abiotic incubations. We reason that they either represent sulfurization of organic matter following microbial sulfate reduction, or were directly synthesized by the native microbial community.

As discussed in the *Introduction*, in the Neumann et al. (2014) experiment, SOC was initially mobilized due to sampling, homogenizing, and/or handling of the sandy aquifer sediment. While such physical perturbations to the subsurface would not occur *in situ*, geochemical perturbations to aquifers can and do occur, and geochemical perturbations hold potential to mobilize organic carbon off sediment into groundwater. For example, changes to pH or ion concentrations could desorb organic carbon while reductive dissolution of carbon containing oxide minerals could release organic carbon into groundwater (Eusterhues et al., 2003; Fontaine et al., 2007; Gu et al., 1994b; Jardine et al., 1989; Kaiser and Zech, 1999; Mikutta et al., 2006). FT-ICR-MS and DOC concentration data (Figures 3–4, S2–S3) signify SOC was abiotically released from aquifer sediment during incubation. Release reflected desorption during equilibration of organic carbon between dissolved and sorbed phases. If it were to get mobilized within the anaerobic Bangladeshi aquifer, the characterized pool of SOC would represent a more energetically favorable carbon source than DOC transported into the aquifer with recharge water due to its higher NOSC (Boye et al., 2017; Keiluweit et al., 2016; LaRowe and Van Cappellen, 2011) (Figure 2).

## 4.2 Microbial processing of organic matter in anaerobic environments

The direct link between microbial use and chemical character afforded by this study demonstrated that compound indices do not predict degradability. During incubation, native microbes successfully transformed all compound types (Figures 5 and S5), including those traditionally considered molecularly recalcitrant (e.g., aromatic and condensed aromatic compounds; (Lutzow et al., 2006)) and those currently considered thermodynamically unviable in anaerobic conditions (e.g., NOSC below –0.3; (Boye et al., 2017; Keiluweit et al., 2016)). The former result aligns with current understanding that under the correct conditions, microbes can process all carbon types (Lehmann and Kleber, 2015; Schmidt et al., 2011), but the latter result challenges the emerging idea that reduced carbon is thermodynamically unviable in anaerobic environments (Boye et al., 2017; Keiluweit et al., 2016).

Chemical indices did, however, give insight into rates of compound transformation. In the anaerobic incubation, NOSC controlled the rate of compound transformation. Aromatic CHO compounds with high NOSC were microbially transformed first (Figures 4 and S3), while alkane and olefinic CHO compounds with low NOSC persisted (i.e., were transformed last; Figures 5 and S5). Thus, while low NOSC values did not inhibit compound degradation, they did slow the rate of compound use. The more energetically favorable compounds with high NOSC were targeted first while the less energetically favorable compounds with low NOSC were targeted last. Compound size was also an important indicator. Larger compounds were actively transformed while smaller compounds persisted (Figures 5, S5 and S6).

Overall, these patterns are consistent with current soil science literature indicating that biotic communities eventually degrade and mineralize all forms of organic matter by continually converting larger energy-rich compounds into smaller energy-poor compounds (Lehmann and Kleber, 2015; Lutzow et al., 2006). Conceptually, the continual input and microbial conversion of carbon compounds coupled with the slower degradation rate of reduced carbon compounds in anaerobic conditions explains the apparent persistence of compounds with low NOSC in anaerobic environments (Boye et al., 2017). The explanation implies that degradation of highly reduced carbon compounds is thermodynamically viable in anaerobic conditions, which is in contrast to previous interpretations, but acknowledges the reaction is less energetically favorable than degradation of more oxidized carbon compounds, and thus, reduced carbon compounds are more slowly processed. In contrast to an incubation bottle, in an open system, where fresh carbon is continually supplied, the slower degradation rate of reduced carbon compounds would enable this portion of the organic carbon pool to build up in an anaerobic environment, skewing the NOSC distribution of soluble carbon to lower values. This interpretation aligns results from the incubation experiment, where every carbon type, regardless of NOSC, was transformed (i.e., lost; Figures 5, S5 and S6), with field observations indicating soluble carbon from anoxic sediments have lower NOSC values than that from oxic sediments (Boye et al., 2017).

**Data availability**

Unprocessed FT-ICR-MS data are available at: https://doi.pangaea.de/10.1594/PANGAEA.876660

Pracht, Lara E; Tfaily, Malak M; Ardissono, Robert J; Neumann, Rebecca B (2017): FT-ICR-MS characterization of organic matter and further sample details of Bangladeshi aquifer sediment incubated with aquifer recharge waters. Dataset #876660 (DOI registration in progress)

**Supplement**

Four tables, six figures and one section providing data and information supporting statements herein.

**Author contributions**

L.E. Pracht and R.B. Neumann designed the incubation experiment. L.E. Pracht and R.J. Ardissono carried out the incubation experiment. M.M. Tfaily conducted the FT-ICR-MS analysis. L.E. Pracht, R.J. Ardissono, M.M. Tfaily and R.B. Neumann processed the FT-ICR-MS data. L.E. Pracht and R.B. Neumann interpreted the FT-ICR-MS data and wrote the manuscript with editing input from M.M. Tfaily and R.J. Ardissono.

**Competing interests**

The authors declare they have no conflict of interest.

## Acknowledgements

We thank Rosalie K. Chu for analytical assistance with FT-ICR-MS, Jesse C. Turner for help with figure formatting, and Steve J. Burges for helpful comments that improved the manuscript. A portion of the research was performed using EMSL, a DOE Office of Science User Facility sponsored by the Office of Biological and Environmental Research (BER) and located at Pacific Northwest National Laboratory.

## Figure Captions

**Figure 1:** Conceptual cartoon indicating where Neumann et al. (2014) collected aquifer sediment (marked with black circle) and aquifer recharge water used in the incubation experiment. Water vacuum As described in the text, recharge water was collected from beneath a rice field and from the sediments of a pond. At the field site, water from rice fields and ponds gets pulled into the aquifer to a depth of ~30-m where irrigation wells are screened. The cartoon is modified from Neumann et al. (2010).

**Figure 2:** Chemical characteristics of mobilized SOC and of DOC in pond and rice field recharge water. (a) Percent of identified compounds in heteroatom groups. Proportional distributions of (b) H-to-C ratios, (c) O-to-C ratios, (d) peak mass, (e) double bond equivalents, and (f) nominal oxidation states of carbon. Percent of identified compounds in groupings based on (g) aromaticity index (alkanes, AI=0; olefinics, 0<AI≤0.5; aromatics, AI>0.5; condensed aromatics, AI≥0.67) and (h) compound classifications (Table S4). Compound classifications are arranged in increases order based on NOSC, as obtained from Boye et al. (2017).

**Figure 3: (a-d)** Average concentrations of DOC and **(e-h)** number of organic compounds in incubation water 1, 18 and 85 days after initiation of the incubation experiment. In panels a-d error bars represent standard deviation around the mean (Table S2). In panels e-h, black circles indicate the total number of compounds present at a given time point. Bars between time points indicate the number of compounds that were identified at both time points, i.e., common (black), identified at the second time point but not at the first, i.e., newly detected (dark grey), and identified at the first time point but not at the second, i.e., lost (light grey).

**Figure 4:** Relative change in chemical indices between day 1 and day 18 for rice field recharge water incubated with native sediment and sterilized sediment. Plotted are the cumulative fractions of compounds that were identified at both time points, i.e., common (black), identified at the second time point but not at the first, i.e., newly detected (dark grey), and identified at the first time point but not at the second, i.e., lost (light grey). Compounds were separated by heteroatom group (CHO and CHO+S groups are shown) and characterized based on H-to-C ratio, O-to-C ratio, S-to-C ratio, nominal oxidation state of carbon (NOSC), peak mass, and aromaticity index.

**Figure 5:** Relative change in chemical indices between day 18 and day 85 for rice field recharge water incubated with native sediment and sterilized sediment. Plotted are the cumulative fractions of compounds that were identified at both time points, i.e., common (black), identified at the second time point but not at the first, i.e., newly detected (dark grey), and identified at the first time point but not at the second, i.e., lost (light grey). Compounds were separated by heteroatom group (CHO and CHO+S groups are shown) and characterized based on H-to-C ratio, O-to-C ratio, S-to-C ratio, nominal oxidation state of carbon (NOSC), peak mass, and aromaticity index.

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

**Table 1. Chemical Character of Initial Waters used in Neumann et al. (2014a) Incubation[*]**

| | Pond Recharge | Rice Field Recharge | Sediment Porewater |
|---|---|---|---|
| % CHO | 26.7 | 22.0 | 9.6 |
| % CHON | 9.9 | 7.3 | 17.6 |
| % CHO+P | 16.0 | 11.8 | 21.3 |
| % CHO+S | 47.3 | 58.9 | 51.4 |
| H-to-C ratio; median, mean (stdev) | 1.33, 1.238 (0.558) | 1.18, 1.100 (0.614) | 0.92, 1.019 (0.528) |
| O-to-C ratio; median, mean (stdev) | 0.286, 0.354 (0.279) | 0.333, 0.404 (0.301) | 0.375, 0.435 (0.299) |
| DBE; median, mean (stdev) | 7.5, 8.4 (5.0) | 8.0, 8.7 (5.0) | 13.0, 13.3 (7.1) |
| NOSC; median, mean (stdev) | -0.56, -0.23 (1.33) | -0.32, 0.14 (1.56) | 0.10, 0.07 (0.86) |
| Peak Mass (Da); median, mean (stdev) | 341.1, 379.6 (155.6) | 352.9, 391.3 (162.4) | 437.0, 454.6 (134.5) |
| % Compounds >600 Da | 6 | 8 | 14 |
| Aromaticity Index | | | |
| % Alkane | 30.9 | 28.6 | 26.5 |
| % Olefinic | 43.7 | 35.3 | 28.6 |
| % Aromatic | 6.5 | 6.6 | 15.5 |
| % Condensed Aromatic | 19.0 | 29.6 | 29.5 |
| Compound Classification | | | |
| % Lipid-like | 14.6 | 14.5 | 3.9 |
| % Protein-like | 15.4 | 12.1 | 5.8 |
| % Amino Sugar-like | 1.8 | 3.0 | 3.2 |
| % Cellulose-like | 3.8 | 2.2 | 4.3 |
| % Lignin-like | 18.3 | 17.1 | 16.9 |
| % Tannin-like | 1.9 | 2.1 | 7.4 |
| % Condensed Hydrocarbon-like | 24.1 | 25.7 | 38.0 |
| % Other | 20.0 | 23.3 | 20.4 |
| % Terrestrially derived (cellulose, lignin, tannin-like) | 24.0 | 21.4 | 28.6 |
| % Microbially derived (lipid, protein, amino sugar-like) | 31.8 | 29.6 | 12.9 |

*percentages reflect only those compounds detected by the FT-ICR-MS analysis method.

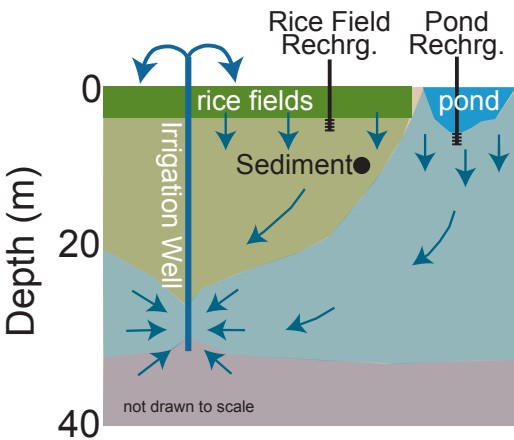

 FIGURE 1

Conceptual cartoon indicating where Neumann et al. (2014) collected aquifer
sediment (marked with black circle) and aquifer recharge water used in the
incubation experiment. As described in the text, recharge water was collected
from beneath a rice field and from the sediments of a pond. At the field site,
water from rice fields and ponds gets pulled into the aquifer to a depth of
~30-m where irrigation wells are screened. The cartoon is modified from
Neumann et al. (2010).

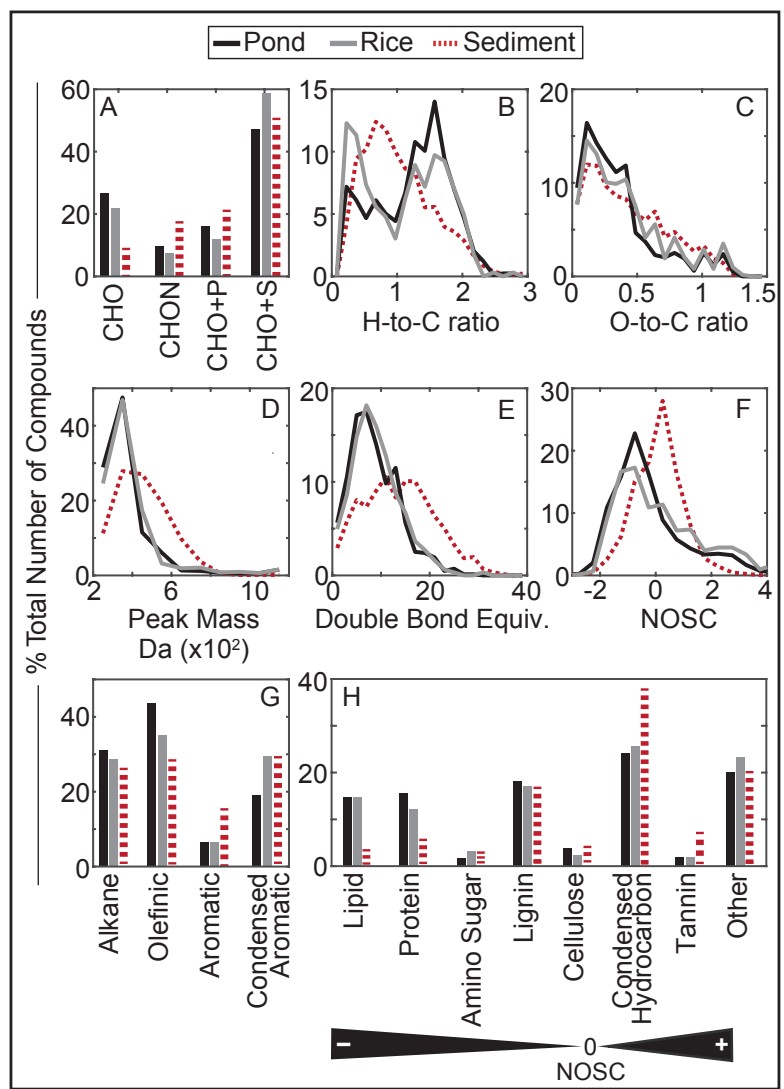

FIGURE 2

Chemical characteristics of mobilized SOC and of DOC in pond and rice field recharge water. (a) Percent of identified compounds in heteroatom groups. Proportional distributions of (b) H-to-C ratios, (c) O-to-C ratios, (d) peak mass, (e) double bond equivalents, and (f) nominal oxidation states of carbon. Percent of identified compounds in groupings based on (g) aromaticity index (alkanes, AI=0; olefinics, 0<AI≤0.5; aromatics, AI>0.5; condensed aromatics, AI≥0.67) and (h) compound classifications (Table S4). Compound classifications are arranged in increases order based on NOSC, as obtained from Boye et al. (2017).

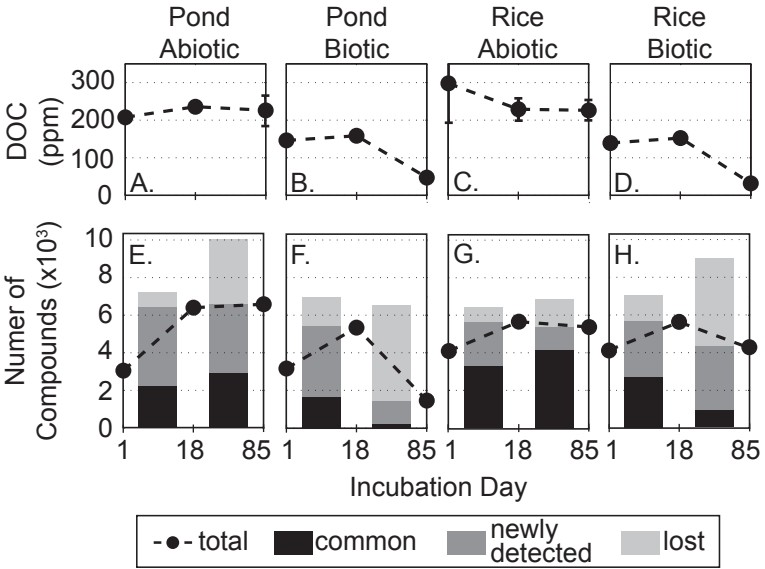

FIGURE 3

(a-d) Average concentrations of DOC and (e-h) number of organic compounds in incubation water 1, 18 and 85 days after initiation of the incubation experiment. In panels a-d error bars represent standard deviation around the mean (Table S2). In panels e-h, black circles indicate the total number of compounds present at a given time point. Bars between time points indicate the number of compounds that were identified at both time points, i.e., common (black), identified at the second time point but not at the first, i.e., newly detected (dark grey), and identified at the first time point but not at the second, i.e., lost (light grey).

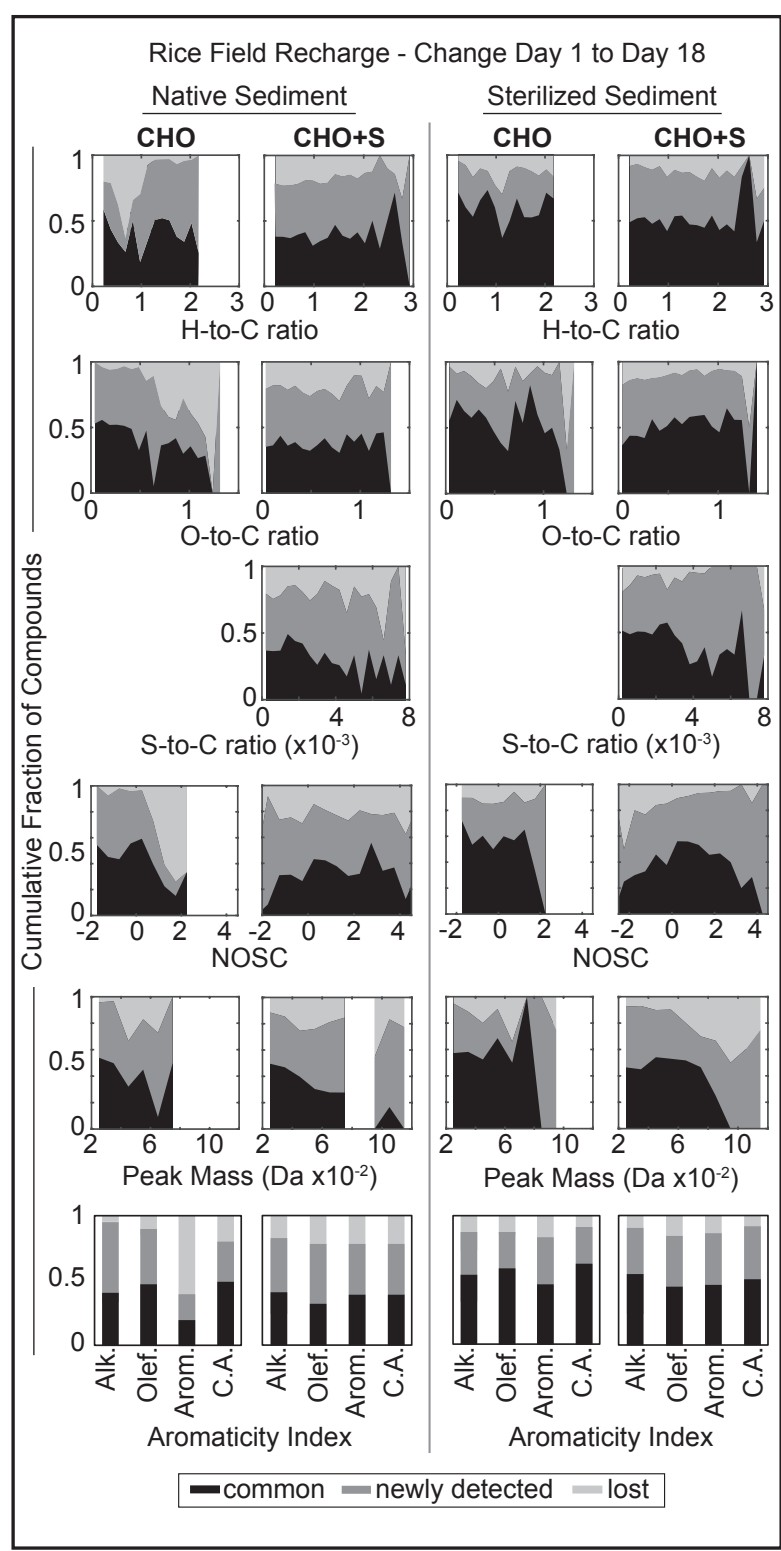

FIGURE 4

Relative change in chemical indices between day 1 and day 18 for rice field recharge water incubated with native sediment and sterilized sediment. Plotted are the cumulative fractions of compounds that were identified at both time points, i.e., common (black), identified at the second time point but not at the first, i.e., newly detected (dark grey), and identified at the first time point but not at the second, i.e., lost (light grey). Compounds were separated by heteroatom group (CHO and CHO+S groups are shown) and characterized based on H-to-C ratio, O-to-C ratio, S-to-C ratio, nominal oxidation state of carbon (NOSC), peak mass, and aromaticity index.

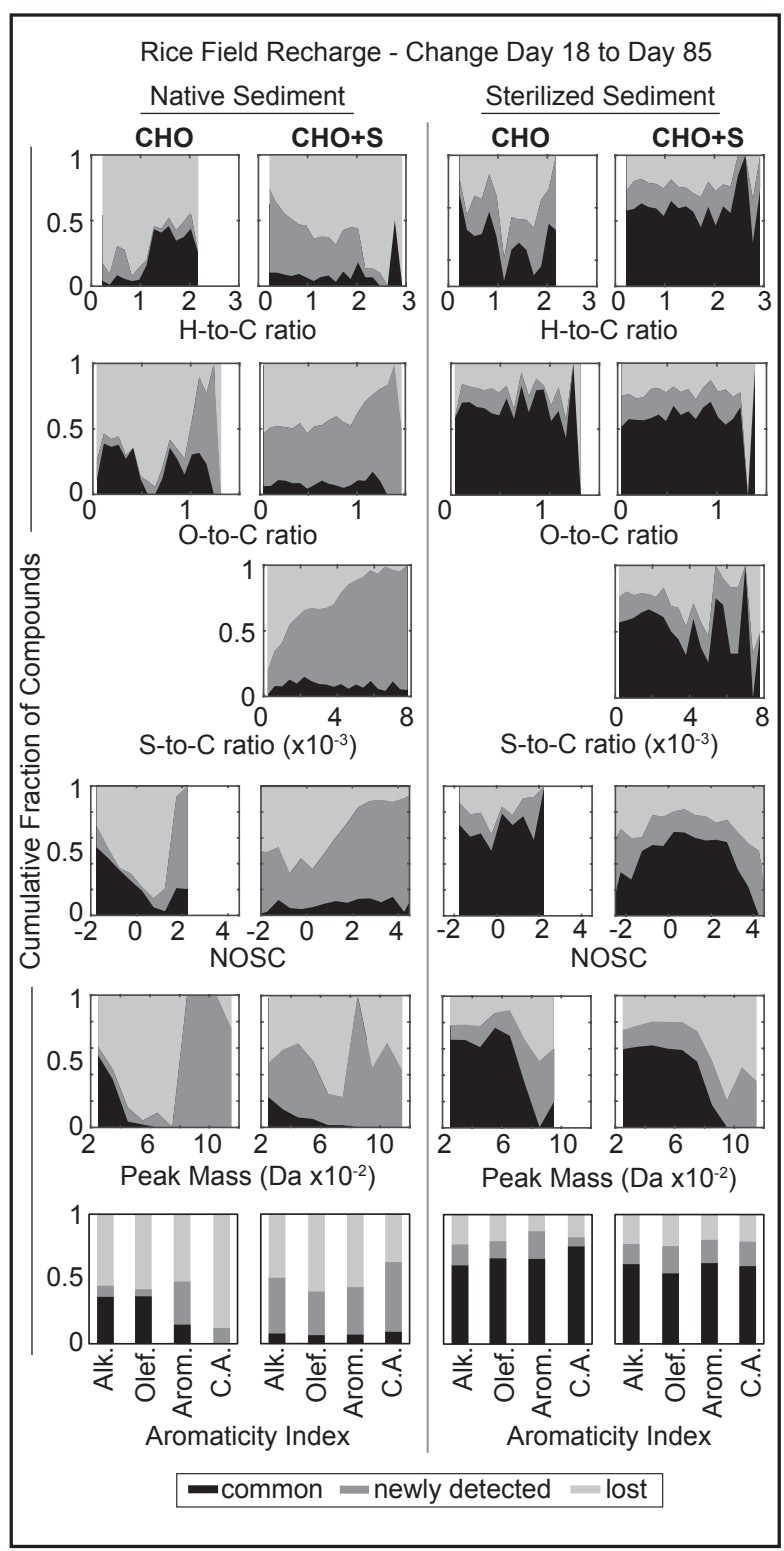

FIGURE 5

Relative change in chemical indices between day 18 and day 85 for rice field recharge water incubated with native sediment and sterilized sediment. Plotted are the cumulative fractions of compounds that were identified at both time points, i.e., common (black), identified at the second time point but not at the first, i.e., newly detected (dark grey), and identified at the first time point but not at the second, i.e., lost (light grey). Compounds were separated by heteroatom group (CHO and CHO+S groups are shown) and characterized based on H-to-C ratio, O-to-C ratio, S-to-C ratio, nominal oxidation state of carbon (NOSC), peak mass, and aromaticity index.