# Peer review of "Molecular characterization of organic matter mobilized from Bangladeshi aquifer sediment: tracking carbon compositional change during microbial utilization"

_Biogeosciences, 2017_

## Referee Comment (RC1) · Anonymous Referee #1 · 28 Jul 2017

The terms and samples collected are unclear to me.

1. Page 2. The term aquifer sediments is in my opinion a misnomer. A lake can have sediments, or an ocean, but I don't see how an aquifer can have sediments. I presume that this is an unconsolidated matrix aquifer consisting of sand and clay particles they must have been deposited by water. But I don't think it is accurate to call the aquifer matrix sediments any more. That said, this aquifer needs to be described better, please.

[Figure]

2. Page 2 and 3. What is aquifer-recharge water? How do you collect pond recharge water and rice field recharge water? (page 3). Please define and describe these terms.

3. How does recharge water (pond or rice field) differ from sediment pore water? I am sorry but I find this confusing.

4. What in reference to these terms above is mobilized SOC?

5. Page 2. How do you know that it's the mobilized SOC that is reacting and not the SOC in place? And if you don't use the terms sediments any more can you call this aquifer matrix organic material?

6. OC degradation proceeds from large molecules to smaller ones and then to methane and CO2. page 2. Do you contradict this later saying that larger molecules are more reactive and smaller ones accumulate?

7. page 3 top of page. In addition to what? On the bottom of the last page you didn't describe what you were doing in the field?

8. page 3, line 5. really? None of the DOC from rice field recharge (what's that) reacted?

9. page 3 line 22. Did these samples come from Neumann's 2014 incubation? How were they preserved all that time? Or was another similar incubation done?

10. page 2, line 19. aquifer recharge waters. Define how you collect this?

11. page 6. line 30. Concentration and character of OC mobilized off aquifer sediment into sediment porewater differed from DOC in aquifer recharge. Do you mean DOC that comes off the aquifer matrix material relative to that from ponds and lakes flowing down? How old is the aquifer matrix material? How old is this aquifer? Wouldn't you expect this?

12. I don't understand how the numbers relate to the pools in line 31 page 6 to line 1 page 7.

13. How do collect mobilized SOC? Is this different from sediment porewater?

14. The figures are awfully small and dense. It would help tremendously if the figure captions were on the same page as the figures. Why is this not allowed?

15. I am uncertain what the x axis are in Fig. 1 B, C, and E.

16. I am uncertain what the x axis is in fig 3, -→ 0, 1, 2, 3??? 0, 1? I thoguth these were day 1 to 18? The figure caption may be accurate but it is too dense.

17. fig. 4, see comment 16.

18. Conclusions. Lines 15-20, page 11. I don't follow this, seems like a lot of methane is being formed in this aquifer already. What are you warning would happen if this pool of SOC was destabilized in situ? And what does this mean, to destabilize it in situ?

Overall, pretty confusing paper, confusing terms, with confusing figures.

---

## Referee Comment (RC2) · Anonymous Referee #2 · 10 Aug 2017

GENERAL COMMENTS

The present study builds on an earlier paper by the group which examined microbial utilization of sedimentary organic carbon upon disturbance (physical homogenization). Here they utilize high resolution mass spectrometry to determine the organic carbon chemistry during microbial processing of the carbon. There are many things to like about the study. Expanding the science of organic carbon in sediments and soils is of importance of understanding the processing (and inversely, storage) of carbon within the subsurface, how it impacts water quality (chemically and biologically), and potential

impacts on atmospheric gas exchange. The mass spectral technique utilized within the study offer a means of examining organic carbon with unprecedented resolution, although there are many caveats that the authors should be careful of and be cautious with their conclusions, particularly without secondary supporting techniques. The physical displacement of the organic carbon provide great insight into what is essentially hiding in the sediments, unavailable for microbial decomposition. Following the displacement with incubation studies helps to confirm that these compounds are decomposable, but not in the physically unaltered sediments. This is an important insight that should be the highlight of the manuscript.

The downside of the manuscript, as it presently stands, is that the primary conclusion, and, in fact, the central theme, is that carbon displaced by physical homogenization will lead to its decomposition by microbial action. There is just no way for me to conceive how sediments 9-m below the ground surface could be homogenized, and making this a central theme of the manuscript is highly problematic. However, by reversing the concept and considering this an examination of the carbon stored in the sediments, the study not only moves to solid ground (no pun intended), but the displacement and subsequent microbial utilization of the sediment organic carbon becomes a highly clever means to illuminate the chemistry of the compounds. I would therefore strongly suggest that the authors consider inverting their view of the central theme and making the paper about the chemistry of organic carbon within the Holocene aquifer of Bangladesh.

One final general comment is that there seems to be a number of locations in the manuscript where microbial processing of organic carbon is not correctly portrayed. I would therefore also suggest the authors seek to ensure microbial and biochemical accuracy (possibly consulting with a microbiologist).

DETAILS

Lines 17-18: Organic carbon is not fermented into methane. It's at two (to three) step

process where first the organic carbon (that less than 600 Da) is bacterially fermented (sometimes with a secondary fermentation step), then the metabolites are respired by archaea to methane (methanogenesis).

Lines 19-20: Stick with abiotic sulfidization; as noted below, the methanogen pathway is not warranted here.

Line 22: Here and elsewhere, the term or inference of "recalcitrance" needs to be clarified. As noted in citation later in the manuscript (Schmidt et al, 2011; Lehmann and Kleber, 2015), "molecular recalcitrance" has become an antiquated notion and should be placed in the correct context here. In short, the new realization is that recalcitrance is an ecosystem specific feature. As summarized in Schmidt et al. (2011), lignin and other complex aromatic structures degrade at rates not dissimilar to many starches.

Lines 10-11: Again, the inference to recalcitrance needs to be clarified (or avoided).

Lines 22-23: same comment as for lines 17-18 on page 1.

Line 30: Exoenzymes are indeeded needed for depolymerization but not necessarily to monomers. The size cutoff is the critical factor and is largely considered in the 600 Da range (as noted in the manuscript).

Line 7: The analysis provides insight into the chemistry of the DOC; it does not, however, translate to the bioavailability, which is a far more complex pathway that varies between specific organisms. Even bioaccessibility is not really tracked here—possibly the lack of bioaccessibility is obtained.

Lines 16-17: I think this statement of "aquifer sediments" is splitting hairs. There is an abundance of work done on marine sediments that should be noted, and which could help guide the authors to stronger conclusions, and an extensive body of literature

on soils is also available. For marine sediment analogies, Hedges and co-workers, for example, have done extensive work that has been, in my opinion, groundbreaking and could help with the interpretation here. Also, two recent papers using FT-ICR-MS (Bailey et al., Soil Biol Biochem. 2017, 107: 133-; Boye et al., Nature Geosci. 2017, 10, 415-), in which one of the authors was involved, could also be helpful and describe organic carbon physically isolated in soils and thermodynamically protected in terrestrial sediments.

Lines 22-32: How long were the samples stored? And at what temperature?

Line 24: Homogenization of the sediments alters that physically accessibility massively. As noted in Bailey et al. (Soil Biol Biochem. 2017, 107: 133-), the chemistry of organic carbon changes with pore-size. Adding in the displacement of physically occluded organic carbon, and the system has been changed massively. This can be used to the authors advantage, but a realization that no disturbance can release this proportion of the OC needs to be made. Rather, it provides an ability to access what is not being processed by the microbiota.

Lines 17-: What proportion of the mass spec data was successfully assigned? Or, in other words, what proportion of the mass spectral data were left unassigned. This is a really important aspect of the analysis when (semi-) quantification is being attempted.

Line 25: This is already an outdated concept. The present manuscript notes this later in the references to Schmidt et al. (2011) and Lehmann and Kleber (2015), and the authors should hold to the updated theory of sediment/soil organic carbon.

Line 22-23: The process described here for microbial metabolism is not correct. Catabolism is the transfer of compounds into useful energy. The authors are correct

that for the organisms to utilize carbon compounds for catabolism or anabolism they must be less than ca. 600 Da. However, to decrease polymer size, extracellular enzymes are (typically) used, and this is independent of catabolism. In fact, catabolic energy is needed to synthesize the molecules, and thus would be an endergonic process.

Line 27-29: As noted in the M-M section, FT-ICR-MS is not quantitative, and thus providing the % differences should be done with caution.

Line 30: Once again, the concept of chemical recalcitrance is now antiquated and should be placed within the ecological context of the environment (see Schmidt et al., 2011, for example).

Page 8 Line 6: "thermodynamically accessible" is not an appropriate description. A reaction is either thermodynamically viable or it is not—there is no middle ground. And accessible is not a term that would equate to viability. Accessible leads one to think of physical access rather than biochemical viability.

Line 25: There is more than fermentation and methanogensis happening here. Dissimilatory sulfate reduction would be taking place and other respiration may as well (DIRB, for example).

Lines 32-33: Are the number of compounds identified meaningful given the noted limitation of the FT-ICR-MS approach?

Line 15-16: I don't understand this statement. If I look at Figure 2 and compare DOC levels for pond water (b) to rice (d) for the biotic incubation, they look exactly the same; for the abiotic incubations, the rice paddy water created great DOC than the pond water, which is the opposite of the text.

Line 4: This is (at least) a two-step process: Fermentation and then methanogenesis.

Line 13: Consider the concepts in these referenced (Lehmann and Kleber, 2015; Schmidt et al., 2011) earlier in the manuscript, and place your findings within the context of current organic matter processing paradigms.

Lines 17-27: This is an excellent paragraph.

Lines 9-10: I would recommend removing the implication of methanogenesis in sulfidization of organic matter. The sulfur enzymes would not be sufficient to provide this signal.

Line 18: Again, remove "fermented into methane". The organic carbon compounds are fermented, and then methanogenesis transpires.

Line 18: The posed "If this pool of SOC were destabilized in situ" is an important point. How would the organic matter be destabilized? In the Neumann et al. (2014) study, the sediments are homogenized, leading to the release of organic matter that is then subject to microbial degradation. It is interesting that the organic matter is rapidly consumed but not surprising. Presumably the OM is physically isolated, or partially mineral protected, and not available for microbial utilization. Residing 9 m below the surface, it is hard to imagine a process that would lead to destabilization. As such, I would recommend moving away from this position and instead focus on the interesting aspect of its chemistry—the metabolism of the OM advances our understanding of their composition and that a protection mechanism must be in play.

Lines 20-26: I don't see the present study supporting the conclusions drawn in this paragraph. The summary provided in lines 27-31 are reasonable and should remain the emphasis. Albeit that I support reasonable speculation, the extension of a homogenized sediment release well-protected organic carbon as an inference into field setting is not warranted. As noted in the comment above, the power of this study is

in characterizing the OM that is protected, not in biogeochemical processes that follow unrealistic release upon physical homogenization.

Line 17: The authors are correct to compare their finding to the soil science literature. It's important to note that there is really a fine line between subsurface soils and sediments. In fact, when soil scientist describe a C horizon (or horizons), they have effectively crossed the boundary into sediments. Top soils that undergo greater input and turnover are a different story, but the subsurface across depths is more similar.

---

## Author Response (AR1)

We the authors were initially requested to submit detailed responses to each reviewer comment, but not to prepare a revised manuscript. After reviewing these initial responses, the editor requested that we revise our manuscript. Below we show each reviewer comment in *blue italic* text, our initial responses to these comments in black un-italicized text, and a description of the corresponding revisions we made to the text to address each reviewer comments in **bold, black italicized** text. At the end of this document is a marked up version of our manuscript that tracks all of the changes made to the manuscript during the revision process.

Anonymous Referee #1:

*The terms and samples collected are unclear to me.*

*1. Page 2. The term aquifer sediments is in my opinion a misnomer. A lake can have sediments, or an ocean, but I don't see how an aquifer can have sediments. I presume that this is an unconsolidated matrix aquifer consisting of sand and clay particles they must have been deposited by water. But I don't think it is accurate to call the aquifer matrix sediments any more.*

> We disagree that the use of "sediment" is incorrect. Basic groundwater textbooks (e.g., Fetter (2001) Applied Hydrogeology, 4th edition, Prentice Hall, Upper Saddle River, New Jersey) use sediment to describe aquifer materials. In the Fetter textbook, sediment is defined as: "assemblages of individual grains that were deposited by water, wind, ice or gravity. There are openings called pore spaces between the sediment grains, so that sediments are not solid." The book goes on to describe porosity and classification of sediments and permeability of sediments as it relates to groundwater aquifers.
> **We kept the use of "sediment" in the revised manuscript.**

*That said, this aquifer needs to be described better, please.*

> The aquifer is well described in previous publications. We will modify section 2.1 to include a better description of the aquifer and we will point the reader towards these references if they desire additional information about the aquifer.
> **We added aquifer information to section 2.1**

*2. Page 2 and 3. What is aquifer-recharge water? How do you collect pond recharge water and rice field recharge water? (page 3). Please define and describe these terms.*
*3. How does recharge water (pond or rice field) differ from sediment pore water? I am sorry but I find this confusing.*

> In response to comments 2 and 3 above, we will include a new figure in the manuscript that shows a cross section of the field site and visually indicates what we are calling recharge water and sediment porewater. This figure will build upon Figure 1d in Neumann et al. (2010) *Nature Geoscience*, which is included below for easy reference:

[Figure]

Aquifer recharge water is water that is flowing into the aquifer from surface water sources, recharging the aquifer, but it has not yet entered the aquifer and thus is not yet technically groundwater. As stated in our manuscript (page 3): "Filtered (0.45 µm) aquifer-recharge waters were collected from underneath a rice field and from pond sediments." We will mark on the figure above where these water samples were collected and we will add information to section 2.1 describing how we collected these water samples.

In contrast to the recharge waters, sediment porewater is water surrounding the collected aquifer sediment, which we vacuum extracted off of the collected sediment back in the laboratory (page 3). We can clarify this point in the added figure.

> ***We added a new figure, Figure 1, which indicates conceptually where incubation materials were collected at the field site. We also added sampling and collection details to section 2.1 and clarified in this section what is meant by sediment porewater and aquifer recharge water.***

*4. What in reference to these terms above is mobilized SOC?*

Mobilized SOC is organic carbon that came off of the aquifer sediment during sampling, homogenization and was dissolved in the sediment porewater. We will add this clarification to section 2.1.
> ***We added this clarification to section 2.1***

*5. Page 2. How do you know that it's the mobilized SOC that is reacting and not the SOC in place?*

As described in the Neumann et al. (2014a) manuscript, the amount of DOC detected in the initial incubation water was fully explained by that detected in sediment porewater before the sediment was used in the incubation. Therefore, the DOC in the incubation water, at least initially, was largely carbon that was in the sediment porewater and thus was carbon that was mobilized off the aquifer matrix. In this previously published experiment, dissolved concentrations of organic carbon then decreased during the incubation and methane was produced. The amount of produced methane was equivalent to the amount of DOC lost during the incubation. These mass balance constraints point to reaction of mobilized SOC.
> ***The mass balance arguments of Neumann et al. (2014) were outlined in section 2.1.***

In the current manuscript, we characterized DOC in the sediment porewater (section 3.2)

and then compare the FT-ICR-MS signature of this carbon to that in the initial incubation water (section 3.3). Via this comparison, we similarly conclude that most of the carbon in the initial incubation water was that mobilized from the aquifer sediment. We then go onto to discuss the possibility that between day 1 and day 18 of the incubation, more SOC was abiotically desorbed off the aquifer sediment into the incubation water.

> *This discussion of abiotic desorption was kept in sections 3.2 and 3.3 and re-iterated in the conclusion section.*

We do not deny the possibility that SOC sorbed to or associated with the aquifer sediment could be processed by microbes – but our study is looking at changes in the water, and thus is necessarily tracking microbial processing of SOC mobilized off the aquifer sediment into the water phase.

*And if you don't use the terms sediments any more can you call this aquifer matrix organic material?*

We intend to keep using the term "sediment."

> *We kept the use of "sediment" in the revised manuscript.*

*6. OC degradation proceeds from large molecules to smaller ones and then to methane and CO2. page 2. Do you contradict this later saying that larger molecules are more reactive and smaller ones accumulate?*

We did not mean to state that small molecules accumulate. Searching our document for the term "accumulate" did not turn up any results. We used the word "persisted" because we are looking at what changes did or did not happen from one time step to the next. The fact that small compounds persisted longer than larger compounds aligns with the pattern of OC degradation proceeding from large molecules to smaller ones and then to methane and $CO_2$. There is no contradiction. We do clarify our use of the word persistence on page 10: "Our interpretation is that persistent compounds were not necessarily biologically inaccessible, but rather they were more slowly processed or were regenerated during degradation of larger compounds."

> *We kept this clarifying sentence in the revised manuscript.*

*7. page 3 top of page. In addition to what? On the bottom of the last page you didn't describe what you were doing in the field?*

We were referring to efforts described previously in the paragraph. In addition to the effort to, "identify the carbon compounds mobilized off aquifer sediment in the Neumann et al. (2014a) experiment and to track chemical compositional changes as SOC was converted into methane," we also characterized DOC in the aquifer-recharge waters. We will modify this paragraph to make this connection more clear.

> *We fully revised and re-wrote the introduction. This sentence does not exist in the revised manuscript.*

*8. page 3, line 5. really? None of the DOC from rice field recharge (what's that) reacted?*

Yes. These are peer-reviewed results from Neumann et al. (2010) *Nature Geoscience*. This result was described in the introduction of this manuscript to provide context for the FT-ICR-MS study and results.

> *We fully revised and re-wrote the introduction. This sentence does not exist in the revised manuscript*

*9. page 3 line 22. Did these samples come from Neumann's 2014 incubation? How were they preserved all that time? Or was another similar incubation done?*

Yes. The samples came from the Neumann et al. (2014) incubation. In section 2.2 we note how the samples were preserved: "Aliquots (100–200 µL) of vacuum-filtered water samples from destructively sampled incubation bottles were immediately 5 frozen and later packed in dry ice for shipment to the Environmental Molecular Sciences Laboratory (EMSL) at Pacific Northwest National Laboratory in Richland, WA." We will modify this sentence to clarify that the analyzed samples came from the Neumann et al. (2014) incubation and that they were filtered and kept frozen until analysis by FT-ICR-MS.

**We added clarifying information on the samples analyzed by FT-ICR-MS in both section 2.1 and section 2.2.**

*10. page 2, line 19. aquifer recharge waters. Define how you collect this?*

See our response to comments #2 and #3. We will clarify how we collected aquifer recharge waters in the method section.

**We added this clarifying information to section 2.1**

*11. page 6. line 30. Concentration and character of OC mobilized off aquifer sediment into sediment porewater differed from DOC in aquifer recharge. Do you mean DOC that comes off the aquifer matrix material relative to that from ponds and lakes flowing down? How old is the aquifer matrix material? How old is this aquifer? Wouldn't you expect this?*

Yes. We are comparing FT-ICR-MS signature of carbon that came off the aquifer sediment to DOC found in the porewater of surface water sources that recharge the aquifer. We agree it is not surprising that the two carbon sources differ. One is presumably a young carbon source (the recharge water) and one is presumably an older carbon source (the aquifer sediment). Therefore, we believe there is value in comparing the two. The sentence at the start of the paragraph sets up this comparison, it is not meant to reflect surprise at this finding.

The aquifer is ~5000 years old. We will add this information to section 2.1 when we describe the aquifer in greater detail (see response to comment #1)

**We added this clarifying information to section 2.1**

*12. I don't understand how the numbers relate to the pools in line 31 page 6 to line 1 page 7.*

The numbers in the parentheses (line 1 page 7) are comparing DOC concentrations in sediment porewater to DOC concentrations in the two recharge waters. We will modify this sentence to make this comparison more clear.

**We modified the text to read: "Sediment porewater, which was shown by Neumann et al. (2014) to have a higher dissolved organic carbon concentration than recharge waters (1059 ± 186 mg/L vs. 17 ± 7 mg/L in rice field recharge water and 30 ± 3 mg/L in pond recharge water), also had a higher total number of assigned formulas (5263 vs. 627 in rice field recharge water and 835 in pond recharge water)."**

*13. How do collect mobilized SOC? Is this different from sediment porewater?*

Dissolved organic carbon in sediment porewater (i.e., the water that was vacuum extracted off the sediment) is what we refer to as mobilized SOC. As stated in our response to comments #2 and #3, we will clarify this point in section 2.1 and in the new figure we will add.

**We modified the text to read: "Neumann et al. (2014) collected sediment for the incubation experiment from an aquifer depth of 9.1 m (Figure 1), where groundwater arsenic and methane concentrations were relatively low (~6 µg/L and ~5 mg/L, respectively). They homogenized the sediment and collected sediment porewater (i.e.,**

*water surrounding the aquifer sediment) by vacuum filtering (0.2 μm) multiple sediment aliquots. Dissolved organic carbon (DOC) concentrations in sediment porewater ranged between 856 and 1219 mg/L (Neumann et al., 2014) (SI Table S1), which was much higher than ~4 mg/L DOC concentration measured in groundwater at the field site (Swartz et al., 2004). Neumann et al. (2014) concluded that organic carbon was mobilized off the sediment into porewater during sampling and/or homogenization of the sediment. In this manuscript, we refer to organic carbon dissolved in the Neumann et al. (2014) sediment porewater as 'mobilized SOC.'*"

*14. The figures are awfully small and dense. It would help tremendously if the figure captions were on the same page as the figures. Why is this not allowed?*

See reply to comments #15 - #17 – we will modify the figures, increasing their size to facilitate easier axis labeling. As for the figure captions, the issue of them not being on the same page as the figure is isolated to the review process. If accepted, captions will be directly below figures in the published manuscript version.

**We increased the size of the figures, including the font in the figures. We added the figure captions to the bottom of the figures.**

*15. I am uncertain what the x axis are in Fig. 1 B, C, and E.*

The x-axis values are for the corresponding ratios given at the top of each plot. We will modify the figure and move the names from the top of each plot to under the x-axis to reduce confusion.

**We modified the figures as stated.**

*16. I am uncertain what the xaxis is in fig3,-→0,1,2,3??? 0,1? I thoguth these were day 1 to 18? The figure caption may be accurate but it is too dense.*
*17. fig. 4, see comment 16.*

As was the case for comment #15 – the x-axis values are for the corresponding ratios given at the top of each plot. We will modify the figure and move the names from the top of each plot to under the x-axis to reduce confusion.

**We modified the figures as stated.**

*18. Conclusions. Lines 15-20, page 11. I don't follow this, seems like a lot of methane is being formed in this aquifer already. What are you warning would happen if this pool of SOC was destabilized in situ? And what does this mean, to destabilize it in situ?*

We were trying to indicate that in the incubation, the organic carbon associated with the aquifer sediment was bioavailable, and if groundwater chemistry changed at the field site such that it facilitated mobilization of organic carbon off the aquifer sediment into groundwater, within the aquifer (i.e., *in situ*), then it could fuel microbial reactions. We were trying to relate the incubation results back to the field site. However, reviewer #2 also had concerns about this approach. Therefore, during revision we will revise our efforts to connect incubation results back to the field site.

**We revised this section of the conclusions to read: "As discussed in the Introduction, in the Neumann et al. (2014) experiment, SOC was initially mobilized due to sampling, homogenizing, and/or handling of the sandy aquifer sediment. While such physical perturbations to the subsurface would not occur in situ, geochemical perturbations to aquifers can and do occur, and geochemical perturbations hold potential to mobilize organic carbon off sediment into groundwater. For example, changes to pH or ion concentrations could desorb organic carbon while reductive dissolution of carbon containing oxide minerals could release organic carbon into groundwater (Eusterhues et al., 2003; Fontaine et al., 2007; Gu et al., 1994b; Jardine**

*et al., 1989; Kaiser and Zech, 1999; Mikutta et al., 2006). FT-ICR-MS and DOC concentration data (Figures 3–4, S2–S3) signify SOC was abiotically released from aquifer sediment during incubation. Release reflected desorption during equilibration of organic carbon between dissolved and sorbed phases. If it were to get mobilized within the anaerobic Bangladeshi aquifer, the characterized pool of SOC would represent a more energetically favorable carbon source than DOC transported into the aquifer with recharge water due to its higher NOSC (Boye et al., 2017; Keiluweit et al., 2016; LaRowe and Van Cappellen, 2011) (Figure 2)."*

*Overall, pretty confusing paper, confusing terms, with confusing figures.*

We are confident that an added overview figure and text in methods section 2.1 will clarify our terms (i.e., mobilized SOC, aquifer recharge water), which was clearly a large contributor to reviewer confusion.

Anonymous Referee #2:

*GENERAL COMMENTS*
*The present study builds on an earlier paper by the group which examined microbial utilization of sedimentary organic carbon upon disturbance (physical homogenization). Here they utilize high resolution mass spectrometry to determine the organic carbon chemistry during microbial processing of the carbon. There are many things to like about the study. Expanding the science of organic carbon in sediments and soils is of importance of understanding the processing (and inversely, storage) of carbon within the subsurface, how it impacts water quality (chemically and biologically), and potential impacts on atmospheric gas exchange. The mass spectral technique utilized within the study offer a means of examining organic carbon with unprecedented resolution, although there are many caveats that the authors should be careful of and be cautious with their conclusions, particularly without secondary supporting techniques. The physical displacement of the organic carbon provide great insight into what is essen- tially hiding in the sediments, unavailable for microbial decomposition. Following the displacement with incubation studies helps to confirm that these compounds are de- composable, but not in the physically unaltered sediments. This is an important insight that should be the highlight of the manuscript.*

We appreciate the positive feedback and the constructive suggestions.

*The downside of the manuscript, as it presently stands, is that the primary conclusion, and, in fact, the central theme, is that carbon displaced by physical homogenization will lead to its decomposition by microbial action. There is just no way for me to conceive how sediments 9-m below the ground surface could be homogenized, and making this a central theme of the manuscript is highly problematic. However, by reversing the concept and considering this an examination of the carbon stored in the sediments, the study not only moves to solid ground (no pun intended), but the displacement and sub- sequent microbial utilization of the sediment organic carbon becomes a highly clever means to illuminate the chemistry of the compounds. I would therefore strongly suggest that the authors consider inverting their view of the central theme and making the paper about the chemistry of organic carbon within the Holocene aquifer of Bangladesh.*

We did not intend for our primary conclusion to be that physical homogenization of aquifer sediment 9-m below the ground surface will lead to carbon displacement and microbial decomposition. We agree that there is no conceivable way for sediments at this depth to experience physical homogenization. We intended for our manuscript to be about the chemistry and bioavailability of organic carbon within the studied aquifer in Bangladesh. We were trying to make the argument that the organic carbon studied in the incubation experiment was carbon that was inaccessible in the aquifer, but if subsurface

chemistry changed such that it changed abiotic sorption of carbon to the sediment, or it fueled reductive dissolution of iron oxides that were associated with organic carbon, some portion of this previously inaccessible carbon could get mobilized into the groundwater and become available to microbes. Once available to microbes, the organic carbon could then fuel further reactions. Given the confusion over this point, we will modify the manuscript to make this argument more clear while also noting that homogenization is a large perturbation that likely cannot be fully replicated by the above mentioned *in situ* perturbations, and we will strengthen the theme of organic carbon chemistry in the Holocene aquifer of Bangladesh, as suggested by the reviewer.

> **We fully revised the re-wrote the introduction as well as the conclusion of the manuscript. We believe these revised sections more clearly present our ideas and arguments surrounding aquifer perturbations and they strengthen the theme of SOC chemistry in the Bangladeshi aquifer.**

*One final general comment is that there seems to be a number of locations in the manuscript where microbial processing of organic carbon is not correctly portrayed. I would therefore also suggest the authors seek to ensure microbial and biochemical accuracy (possibly consulting with a microbiologist).*

In the manuscript we used "fermentation of organic carbon into methane," to encompass the entire process of anaerobic methane production. We recognize that methane production is a multi-step process and that fermentation is the first step and that the microbes that actually produce methane from acetate or hydrogen are not the same set that are fermenting organic carbon. We will fix this phrasing to be more technically correct. In addition, we have reached out to a microbiologist.

> **We fixed the phrasing throughout the manuscript.**

*DETAILS*

*Page 1*

*Lines 17-18: Organic carbon is not fermented into methane. It's at two (to three) step process where first the organic carbon (that less than 600 Da) is bacterially fermented (sometimes with a secondary fermentation step), then the metabolites are respired by archaea to methane (methanogenesis).*

See our response above. We are well aware of the multiple steps involved with anaerobic methane production. We were using this phrasing to encompass the entire process. We will modify this phrasing in the manuscript to be more technically correct.

> **We fixed the phrasing to read: "mobilized SOC was converted into methane".**

*Lines 19-20: Stick with abiotic sulfidization; as noted below, the methanogen pathway is not warranted here.*

After consulting with a microbiologist, we have decided to broaden the possible pathways for the formation of organo-sulfur compounds to include anabolism in general (i.e., microbial formation of orano-sulfur compounds for any purpose, not just methanogenesis) and sulfate reduction leading to abiotic sulfidization.

> **The text now reads: "We reason that these detected compounds formed abiotically following microbial reduction of sulfate to sulfide, which could have occurred during incubation but was not directly measured, or they were microbially synthesized."**

*Line 22: Here and elsewhere, the term or inference of "recalcitrance" needs to be clarified. As*

*noted in citation later in the manuscript (Schmidt et al, 2011; Lehmann and Kleber, 2015), "molecular recalcitrance" has become an antiquated notion and should be placed in the correct context here. In short, the new realization is that recalcitrance is an ecosystem specific feature. As summarized in Schmidt et al. (2011), lignin and other complex aromatic structures degrade at rates not dissimilar to many starches.*

We are aware of the ecosystem specific nature of recalcitrance and reference Schmidt et al. (2011) in the manuscript. However, based on responses at conferences and other meetings, there are still many researchers that hold onto the idea of molecular recalcitrance. We were trying to highlight that molecular recalcitrance did not control carbon use in the experiment, which aligns with this emerging understanding that all carbon types are molecularly available. We will add a sentence to the abstract to help make this connection to and alignment with current understanding more clear (i.e., clarify and contextualize our use of molecular recalcitrance as requested by the reviewer).

***Throughout the revised manuscript we clarify that the idea of molecular recalcitrance is outdated, and that our results align with current understanding that microbes can process all carbon types.***

*Lines 10-11: Again, the inference to recalcitrance needs to be clarified (or avoided).*

Similar to our response above, we will modify the manuscript to clarify and contextualize our use of molecular recalcitrance in the manuscript, focusing on how this is an outdated but still used concept and highlighting how our results align with emerging understanding of carbon processing in the environment.

***Throughout the revised manuscript we clarify that the idea of molecular recalcitrance is outdated, and that our results align with current understanding that microbes can process all carbon types.***

*Lines 22-23: same comment as for lines 17-18 on page 1.*

See reply to same comment above.

***Throughout the revised manuscript we clarify that the idea of molecular recalcitrance is outdated, and that our results align with current understanding that microbes can process all carbon types.***

*Line 30: Exoenzymes are indeeded needed for depolymerization but not necessarily to monomers. The size cutoff is the critical factor and is largely considered in the 600 Da range (as noted in the manuscript).*

We will remove monomers from this sentence and replace it with the critical ~600Da-size cutoff.

***The introduction was re-written and this section/sentence was cut.***

*Line 7: The analysis provides insight into the chemistry of the DOC; it does not, however, translate to the bioavailability, which is a far more complex pathway that varies between specific organisms. Even bioaccessibility is not really tracked here—possibly the lack of bioaccessibility is obtained.*

We agree that bioavailability is complex and varies between specific organisms. However, the rate of carbon usage during the incubation does give insight into anaerobic

microbial processing of organic matter for the studied system (i.e., the captured microbial community under the conditions of the incubation). We will modify the manuscript to reflect this more nuanced perspective of "bioavailability" and move away from broadly connecting bioavailability with chemical compositional changes tracked in the incubation.

> **In the revised manuscript we clearly state that** *"accessibility and microbial ecology control carbon bioavailability"*

*Lines 16-17: I think this statement of "aquifer sediments" is splitting hairs. There is an abundance of work done on marine sediments that should be noted, and which could help guide the authors to stronger conclusions, and an extensive body of literature on soils is also available. For marine sediment analogies, Hedges and co-workers, for example, have done extensive work that has been, in my opinion, groundbreaking and could help with the interpretation here. Also, two recent papers using FT-ICR- MS (Bailey et al., Soil Biol Biochem. 2017, 107: 133-; Boye et al., Nature Geosci. 2017, 10, 415-), in which one of the authors was involved, could also be helpful and describe organic carbon physically isolated in soils and thermodynamically protected in terrestrial sediments.*

> We appreciate the references and will include them in our revised manuscript.
>> **We fully integrated Boye et al.(2017) and Bailey et al (2017) into the revised manuscript.**

*Lines 22-32: How long were the samples stored? And at what temperature?*

> After collection, water samples were filtered in an anaerobic glove box and transferred to vials that were kept frozen at -18 C for a period two to three months before shipment to EMSL for FT-ICR-MS analysis. We will add this information to the methods section.
>> **This information was added to section 2.2.**

*Line 24: Homogenization of the sediments alters that physically accessibility massively. As noted in Bailey et al. (Soil Biol Biochem. 2017, 107: 133-), the chemistry of organic carbon changes with pore-size. Adding in the displacement of physically occluded organic carbon, and the system has been changed massively. This can be used to the authors advantage, but a realization that no disturbance can release this proportion of the OC needs to be made. Rather, it provides an ability to access what is not being processed by the microbiota.*

> We appreciate the reference and agree with this comment. See our reply to this similar comment in the General Comment section above.
>> **We have revised our introduction section to include this point:** *"While mobilization was an artifact of the experiment, it allowed us to study a pool of sedimentary organic carbon that was highly bioavailable after mobilization and would have otherwise been inaccessible."*

*Page 5*

*Lines 17-: What proportion of the mass spec data was successfully assigned? Or, in other words, what proportion of the mass spectral data were left unassigned. This is a really important aspect of the analysis when (semi-) quantification is being attempted.*

> 43 to 47% of the mass spectra data were successfully assigned. This percentage assignment was consistent for all of the samples and was at a level that we considered good given the samples were not concentrated or otherwise treated prior to analysis. We will add this information to the methods section of the manuscript.
>> **We added this information to section 2.2.**

*Page 6*

*Line 25: This is already an outdated concept. The present manuscript notes this later in the references to Schmidt et al. (2011) and Lehmann and Kleber (2015), and the authors should hold to the updated theory of sediment/soil organic carbon.*

As stated above, we will modify the manuscript to clarify and contextualize our use of molecular recalcitrance in the manuscript, focusing on how this is an outdated but still used concept, and highlighting how our results align with emerging understanding of carbon processing in the environment.

**Throughout the revised manuscript we clarify that the idea of molecular recalcitrance is outdated, and that our results align with current understanding that microbes can process all carbon types.**

*Page 7*

*Line 22-23: The process described here for microbial metabolism is not correct. Catabolism is the transfer of compounds into useful energy. The authors are correct that for the organisms to utilize carbon compounds for catabolism or anabolism they must be less than ca. 600 Da. However, to decrease polymer size, extracellular enzymes are (typically) used, and this is independent of catabolism. In fact, catabolic energy is needed to synthesize the molecules, and thus would be an endergonic process.*

We appreciate the correction. We have consulted with a microbiologist and will modify the language accordingly.

**We fixed this language. The section now simply reads:** *"the dimensions of bacterial porin structures can lead to the exclusion of large organic solutes (e.g. >600Da), requiring extracellular enzymes (exoenzymes) to breakdown these larger molecules before they can be assimilated by microbial cells (Benz and Bauer, 1988; Nikaido and Vaara, 1985; Weiss et al., 1991)"*

*Line 27-29: As noted in the M-M section, FT-ICR-MS is not quantitative, and thus providing the % differences should be done with caution.*

We agree that FT-ICR-MS is not quantitative and we note this in our methods section, however, multiple previous readers of the manuscript requested such comparisons of the generated data. We will clarify in the manuscript when we compare relative compositions that what we are comparing is the composition detected by the analysis method and not necessarily the full composition of the samples. That said, all samples were treated the same and normalized by the total number of peaks in each sample, making such comparisons more robust.

**We added the following text to the start of the results/discussion section:** *"As previously mentioned, ESI FT-ICR-MS is not quantitative because the ionization efficiency of the different organic compounds vary widely during ESI. Thus, comparisons involve only those compounds detected by the instrument, and not the full composition of each sample. However, all samples were treated the same and each sample was normalized by the total number of detected peaks, making such comparisons more robust."*

**We also added a similar clarification directly to Table 1:** *"percentages reflect only those compounds detected by the FT-ICR-MS analysis method."*

*Line 30: Once again, the concept of chemical recalcitrance is now antiquated and should be*

*placed within the ecological context of the environment (see Schmidt et al., 2011, for example).*

As stated above, we will modify the manuscript to clarify and contextualize our use of molecular recalcitrance in the manuscript, focusing on how this is an outdated but still used concept and highlighting how our results align with emerging understanding of carbon processing in the environment.

> ***Throughout the revised manuscript we clarify that the idea of molecular recalcitrance is outdated, and that our results align with current understanding that microbes can process all carbon types.***

*Line 6: "thermodynamically accessible" is not an appropriate description. A reaction is either thermodynamically viable or it is notâ Ť there is no middle ground. And accessible is not a term that would equate to viability. Accessible leads one to think of physical access rather than biochemical viability.*

We agree with the reviewer that a reaction is either thermodynamically viable or not. We were using "thermodynamically accessible" to point to the energetic favorability of a reaction. We will remove this phrasing from the manuscript and instead use "energetically favorable."

> ***The term thermodynamic accessibility was removed from the revised manuscript and replaces with thermodynamic favorability.***

*Line 25: There is more than fermentation and methanogensis happening here. Dissimilatory sulfate reduction would be taking place and other respiration may as well (DIRB, for example).*

Our sentence was trying to indicate the possibility of microbial sulfate reduction with by our statement "microbes … generated sulfide."  We will clarify this point by specifically calling out the reaction by name (dissimilatory sulfate reduction) in the sentence and including to the possibility of other reactions.

> ***The text was revised to: "In the later incubation period, new organosulfur compounds were detected in biotic incubations but not in abiotic incubations, indicating that microbes either generated sulfide (via dissimilatory sulfate reduction) that abiotically reacted with DOM during this time period or they directly synthesized organosulfur compounds (see section 3.4)"***

*Lines 32-33: Are the number of compounds identified meaningful given the noted limitation of the FT-ICR-MS approach?*

The number of detected compounds is traditionally reported in FT-ICR-MS studies. Even though the method is not quantitative, the number of detected compounds provides insight into the complexity of the samples and is sometime correlated with DOC concentration. Because of these relationships, the number of detected compounds facilitates comparisons within and between different studies using FT-ICR-MS.

> ***The following sentence was added to section 3.1: "The number of detected compounds provides insight into the complexity of the samples (Sleighter and Hatcher, 2008). These numbers indicate that mobilized SOC was more complex than DOC in surface recharge water."***

*Line 15-16: I don't understand this statement. If I look at Figure 2 and compare DOC levels for pond water (b) to rice (d) for the biotic incubation, they look exactly the same; for the abiotic incubations, the rice paddy water created great DOC than the pond water, which is the opposite*

*of the text.*

We are referring to concentration changes over time in the different incubations. At the start of the paragraph we state that DOC concentrations did not change during the incubation for bottles with rice-field recharge water (both biotic and abiotic treatments), but there was an increase in DOC for bottles with pond recharge water between day 1 and day 18 in both the biotic and abiotic treatments. We will modify the text to make it clear that we are talking about a change over time and that we are not comparing the treatments to each other.

***The text was simplified to: "If this interpretation is correct, then increases in DOC concentrations between day 1 and day 18 of the incubation can be attributed to desorption of SOC."***

*Line 4: This is (at least) a two-step process: Fermentation and then methanogenesis.*

We acknowledge this point and will modify the manuscript to be more technically correct.
***We fixed this phrasing throughout the manuscript.***

*Line 13: Consider the concepts in these referenced (Lehmann and Kleber, 2015; Schmidt et al., 2011) earlier in the manuscript, and place your findings within the con- text of current organic matter processing paradigms.*

As stated above, we will modify the manuscript to clarify and contextualize our use of molecular recalcitrance in the manuscript.

***Throughout the revised manuscript we clarify that the idea of molecular recalcitrance is outdated, and that our results align with current understanding that microbes can process all carbon types.***

*Lines 17-27: This is an excellent paragraph.*

We appreciate the positive feedback.

*Lines 9-10: I would recommend removing the implication of methanogenesis in sulfidization of organic matter. The sulfur enzymes would not be sufficient to provide this signal.*

As noted in our response to this same comment made for page 1, after consulting with a microbiologist, we have decided to broaden the possible pathways for the formation of organo-sulfur compounds to include anabolism in general (i.e., microbial formation of orano-sulfur compounds for any purpose, not just methanogenesis) and sulfate reduction leading to abiotic sulfidization.

***The section now reads: "We reason that these detected compounds either formed due to microbial sulfate reduction and sulfide generation during this period, which was not directly measured, followed by abiotic sulfurization of organic matter, or they were microbally synthesized. Both pathways are plausible. Sulfurizaiton of organic matter is well documented (Brown, 1986; Heitmann and Blodau, 2006; Kohnen et al., 1991; Urban et al., 1999), and microbes can synthesize a wide array of organic sulfur compounds (Brosnan and Brosnan, 2006; Madigan et al., 2003; Thauer, 1998)."***

*Line 18: Again, remove "fermented into methane". The organic carbon compounds are fermented, and then methanogenesis transpires.*

We acknowledge this point and will modify the manuscript to be more technically correct. **We fixed this phrasing throughout the manuscript.**

*Line 18: The posed "If this pool of SOC were destabilized in situ" is an important point. How would the organic matter be destabilized? In the Neumann et al. (2014) study, the sediments are homogenized, leading to the release of organic matter that is then subject to microbial degradation. It is interesting that the organic matter is rapidly consumed but not surprising. Presumably the OM is physically isolated, or partially mineral protected, and not available for microbial utilization. Residing 9 m below the surface, it is hard to imagine a process that would lead to destabilization. As such, I would recommend moving away from this position and instead focus on the interesting aspect of its chemistry—the metabolism of the OM advances our understanding of their composition and that a protection mechanism must be in play.*

See our response to this comment in the "General Comments" section at the start of the review.

**We clarified our ideas about in situ destabilization in the introduction section:** *"In the Neumann et al. (2014) incubation experiment, SOC was mobilized during sampling and/or homogenization of the aquifer sediment. While this type of physical disturbance to the aquifer matrix could not occur in situ (i.e., within the aquifer), it is plausible that more realistic aquifer perturbations, such as geochemical changes brought about by large-scale groundwater pumping, could mobilize SOC in situ. For example, mineral-associated organic carbon can get mobilized into groundwater if solution pH increases (Gu et al., 1994a; Jardine et al., 1989; Kaiser and Zech, 1999), if concentrations of ions that compete with organic carbon for sorption sites increase (Jardine et al., 1989; Kaiser and Zech, 1999), or if an influx of dissolved organic carbon fuels microbial reactions that target the mineral phase (e.g., reductive dissolution of iron oxide minerals) (Eusterhues et al., 2003; Fontaine et al., 2007; Mikutta et al., 2006)."*

*Lines 20-26: I don't see the present study supporting the conclusions drawn in this paragraph. The summary provided in lines 27-31 are reasonable and should remain the emphasis. Albeit that I support reasonable speculation, the extension of a homogenized sediment release well-protected organic carbon as an inference into field setting is not warranted. As noted in the comment above, the power of this study is in characterizing the OM that is protected, not in biogeochemical processes that follow unrealistic release upon physical homogenization.*

We agree that aquifer sediment will not experience homogenization, and thus the carbon probed in our incubation study may remain protected indefinitely. However, we do think it is reasonable that some of the SOC studied in the experiment could get mobilized into the aquifer due to desorption or reductive dissolution reactions. This possibility is one motivation for probing the protected OM. We will clarify our conceptual mechanisms for possible *in situ* carbon mobilization, while noting that these mechanisms will not access all the carbon made available during homogenization. Similarly, we will de-emphasize this point/conclusion in the manuscript and strength the focus on the chemistry of protected OM.

**We revised the conclusion section to reference in situ destabilization ideas put forward in the Introduction. The section now reads:** *"As discussed in the Introduction, in the Neumann et al. (2014) experiment, SOC was initially mobilized due to sampling, homogenizing, and/or handling of the sandy aquifer sediment. While such physical perturbations to the subsurface would not occur in situ, geochemical perturbations to aquifers can and do occur, and geochemical perturbations hold*

*potential to mobilize organic carbon off sediment into groundwater. For example, changes to pH or ion concentrations could desorb organic carbon while reductive dissolution of carbon containing oxide minerals could release organic carbon into groundwater (Eusterhues et al., 2003; Fontaine et al., 2007; Gu et al., 1994b; Jardine et al., 1989; Kaiser and Zech, 1999; Mikutta et al., 2006). FT-ICR-MS and DOC concentration data (Figures 3–4, S2–S3) signify SOC was abiotically released from aquifer sediment during incubation. Release reflected desorption during equilibration of organic carbon between dissolved and sorbed phases. If it were to get mobilized within the anaerobic Bangladeshi aquifer, the characterized pool of SOC would represent a more energetically favorable carbon source than DOC transported into the aquifer with recharge water due to its higher NOSC (Boye et al., 2017; Keiluweit et al., 2016; LaRowe and Van Cappellen, 2011) (Figure 2)."*

*Page 12*

*Line 17: The authors are correct to compare their finding to the soil science literature. It's important to note that there is really a fine line between subsurface soils and sediments. In fact, when soil scientist describe a C horizon (or horizons), they have effectively crossed the boundary into sediments. Top soils that undergo greater input and turnover are a different story, but the subsurface across depths is more similar.*

We appreciate the positive comment and insight.

[revised manuscript text omitted]

---

## Referee Report (RR1)

[referee-annotated manuscript omitted]

---

## Author Response (AR2)

February 28, 2018

Dr. Jack Middelbury
Associate Editor
Biogeosciences

Dear Dr. Middelbury:

I am excited to submit a revised version of our manuscript, "Molecular characterization of organic matter mobilized from Bangladeshi aquifer sediment: tracking carbon compositional change during microbial utilization." (bg-2017-275). All of the minor revisions requested by Dr. Hatcher were made. Attached to this letter is a version of the manuscript with the changes tracked so that you can easily see the edits that were made.

I appreciate the time and effort you have put into our manuscript.

Sincerely,

Rebecca Neumann
Assistant Professor
Department of Civil and Environmental Engineering
Box 352700
University of Washington
Seattle, WA 98195
Phone: 206-221-2298
Fax: 206-543-1543
Email: rbneum@uw.edu

[revised manuscript text omitted]

---

## Author Response (AR3)

**UNIVERSITY *of* WASHINGTON**

**CIVIL & ENVIRONMENTAL ENGINEERING**

March 2, 2018

Dr. Jack Middelbury
Associate Editor
Biogeosciences

Dear Dr. Middelbury:

I am excited to submit the final version of our manuscript, "Molecular characterization of organic matter mobilized from Bangladeshi aquifer sediment: tracking carbon compositional change during microbial utilization." (bg-2017-275). All of the minor revisions requested by you have been made.

I appreciate the time and effort you have put into our manuscript.

Sincerely,

Rebecca Neumann
Assistant Professor
Department of Civil and Environmental Engineering
Box 352700
University of Washington
Seattle, WA 98195
Phone: 206-221-2298
Fax: 206-543-1543
Email: rbneum@uw.edu